# Low-level mixed-phase clouds in a complex Arctic environment

Rosa Gierens[1], Stefan Kneifel[1], Matthew D. Shupe[2,3], Kerstin Ebell[1], Marion Maturilli[4], and Ulrich Löhnert[1]

[1]Institute for Geophysics and Meteorology, University of Cologne
[2]Cooperative Institute for Research in Environmental Science, University of Colorado
[3]NOAA Earth System Research Laboratory, Physical Science Division
[4]Alfred Wegener Institute Helmholtz Centre for Polar and Marine Research, Potsdam, Germany

**Correspondence:** Rosa Gierens (rgierens@uni-koeln.de)

**Abstract.**

Low-level mixed-phase clouds (MPC) are common in the Arctic. Both local and large scale phenomena influence the properties and lifetime of MPCs. Arctic fjords are characterized by complex terrain and large variations in surface properties. Yet, not many studies have investigated the impact of local boundary layer dynamics and their relative importance on MPCs in the fjord environment. In this work, we used a combination of ground-based remote sensing instruments, surface meteorological observations, radiosoundings, and reanalysis data to study persistent low-level MPCs at Ny-Ålesund, Svalbard, for a 2.5 year period. Methods to identify the cloud regime, surface coupling, as well as regional and local wind patterns were developed. We found that persistent low-level MPCs were most common with westerly winds, and the westerly clouds had a higher mean liquid ($42 \ \mathrm{gm^{-2}}$) and ice water path ($16 \ \mathrm{gm^{-2}}$) compared to those with easterly winds. The increased height and rarity of persistent MPCs with easterly free-tropospheric winds suggest the island and its orography have an influence on the studied clouds. Seasonal variation of the liquid water path was found to be minimal, although the occurrence of persistent MPCs, their height and ice water path all showed notable seasonal dependency. Most of the studied MPCs were decoupled from the surface (63–82 % of the time). The coupled clouds had 41 % higher liquid water path than the fully decoupled ones. Local winds in the fjord were related to the frequency of surface coupling, and we propose that katabatic winds from the glaciers in the vicinity of the station may cause clouds to decouple. We concluded that while the regional to large scale wind direction was important for the persistent MPC occurrence and properties, also the local scale phenomena (local wind patterns in the fjord and surface coupling) had an influence. Moreover, this suggests that local boundary layer processes should be described in models in order to present low-level MPC properties accurately.

## 1 Introduction

The Arctic is warming more rapidly than any other area on Earth due to climate change (Serreze et al., 2009; Solomon et al., 2007; Wendish et al., 2017). It is well established that clouds strongly impact the surface energy budget in the Arctic (Dong et al., 2010; Shupe and Intrieri, 2004), but feedback processes that include clouds are not well characterized (Choi et al., 2014; Kay and Gettelman, 2009; Serreze and Barry, 2011). Particularly low-level mixed-phase clouds are important for the warming

of near-suface air (Shupe and Intrieri, 2004; Intrieri et al., 2002; Zuidema et al., 2005). The multitude of micro-physical and dynamical processes within the cloud and the interactions with local and large scale processes make these mixed-phase clouds difficult to represent in numerical models (Morrison et al., 2008, 2012; Komurcu et al., 2014). Improvements in the process-level understanding are still required to improve the description of low-level mixed-phase clouds in climate models (McCoy et al., 2016; Kay et al., 2016; Klein et al., 2009).

Previous studies have shown the prevalence of mixed-phase clouds (MPC) across the Arctic (Shupe, 2011; Mioche et al., 2015). MPCs occur in every season, with the highest occurrence in autumn and in the lowest 1 km above the surface, and can persist from hours to days (Shupe et al., 2006; Shupe, 2011; De Boer et al., 2009). The persistent low-level MPCs have a typical structure that consists of one or more super-cooled liquid layers embedded in a deeper layer of ice, where liquid is usually found at cloud top and the ice precipitating from the cloud may sublimate before reaching the ground (Morrison et al., 2012, and references therein). Several studies have shown an increase in cloud ice to coincide with increase in cloud liquid, suggesting that ice production is linked to the liquid water in the cloud (Korolev and Isaac, 2003; Shupe et al., 2004, 2008, 2006; Westbrook and Illingworth, 2011; Morrison et al., 2005). The amount of liquid and ice, and the partitioning between the condensed phases (i.e., phase-partitioning), are important parameters due to their key role in determining the clouds' radiative effect (Shupe and Intrieri, 2004).

A variety of environmental conditions can effect cloud micro- and macro-physical properties. According to simulations over different surface types, changes in surface properties lead to changes in the thermodynamic structure of the atmospheric boundary layer, the extent of dynamical coupling of the cloud to the surface, as well as the micro-physical properties of the MPC (Morrison et al., 2008; Li et al., 2017; Savre et al., 2015; Eirund et al., 2019). Also observational evidence on the connection between changes in surface conditions and MPC occurrence (Morrison et al., 2018) as well as thermodynamic structure and droplet number concentration (Young et al., 2016) have been found. Kalesse et al. (2016) discovered in a detailed case study that for the MPC in question, phase partitioning was affected by the coupling of the cloud to the surface, large scale advection of different airmasses as well as local scale dynamics. On the contrary, Sotiropoulou et al. (2014) did not find differences in cloud water properties between coupled and decoupled clouds. Scott and Lubin (2016) show that at Ross Island, Antarctica, orographic lifting of marine air is likely causing thick MPCs with high ice water content. Changes in aerosol population, especially ice nucleating particles (INP), have been found to modulate the ice formation rate (Jackson et al., 2012; Morrison et al., 2008; Norgren et al., 2018; Solomon et al., 2018). To complicate matters further, the cloud also modifies the boundary layer where it resides by modifying radiative fluxes, generating turbulence (due to cloud top cooling), and vertically redistributing moisture (Morrison et al., 2012; Solomon et al., 2014; Brooks et al., 2017).

While being common in the entire Arctic, MPCs are most frequently observed in the area around Svalbard and the Norwegian and Greenland seas (Mioche et al., 2015). Nomokonova et al. (2019b) report one year of ground based remote sensing observations of clouds at Ny-Ålesund, Svalbard, and find that 20 % of the time single-layer MPCs (defined as single-layer clouds with ice an liquid occurring at any height of the cloud) were present, with highest frequency in autumn and in late spring/early summer. Svalbard lies in a region where intrusions of warm and moist air from lower latitudes are common (Woods et al., 2013; Pithan et al., 2018), and differences in air mass properties have been associated with differences in ice and liquid water

content and particle number concentration in MPCs (Mioche et al., 2017). Locally, the archipelago exhibits large variations in surface properties (glaciers, seasonal sea-ice and snow cover) as well as orography. There are less MPCs over the islands than over the surrounding sea during winter and spring, while during summer and autumn the differences are small (Mioche et al., 2015), indicating that the islands modify the local boundary layer and the associated clouds. How the orography influences the low MPCs in more detail is difficult to study using space-born radars due to the rather big blind zone and considerably large footprint. Aircraft and ground-based remote sensing, together with modelling studies, are better suited for answering this question.

In this paper we investigate persistent low-level mixed-phase clouds (P-MPC) observed above Ny-Ålesund on the west coast of Svalbard. Mountainous coastlines are common at Svalbard, Greenland, and elsewhere in the Arctic (Esau and Repina, 2012). Ny-Ålesund is an excellent site to study low-level MPCs in such complex environments. The time period considered is June 2016–October 2018, when a cloud radar of the University of Cologne was operating at the French–German Arctic Research Base AWIPEV as part of the project Arctic Amplification: Climate Relevant Atmospheric and Surface Processes, and Feedback Mechanisms (AC)[3]. A combination of ground based remote sensing instruments, surface meteorological observations, radiosoundings, and reanalysis data was used to identify and characterize the P-MPCs, to describe the extent of surface coupling, and to evaluate these in the context of wind direction in the area around the station. In addition to providing a description of micro- and macro-physical properties of P-MPCs and their seasonal variation, we aim to identify some of the impacts the coastal location and the mountains have on the observed P-MPCs, as well as determine the relevance of surface coupling for cloud properties at the site. In Sect. 2 the measurement site, the instrumentation and data products used are introduced, followed by the description of the methodology developed to identify persistent low-level MPCs (Sect. 3.1), coupling of the cloud to the surface (Sect. 3.2), and the approach to describe regional and local wind conditions (Sect. 3.3 and 3.4). The results and discussion section describes the occurrence of P-MPC (Sect. 4.1) and their average properties as well as variation under different seasons and dynamical conditions. The relationship between P-MPC and the regional wind direction (Sect. 4.2), different seasons (Sect. 4.3), surface coupling (Sect. 4.4) as well as local wind conditions (Sect. 4.5) are considered. The results are discussed in the context of atmospheric temperature and humidity and considering previous studies at Ny-Ålesund and other Arctic sites. In the end the main aspects are summarized followed by conclusions in Sect. 5.

## 2 Observations

### 2.1 Measurement site

The measurements were carried out at the French–German Arctic Research Base AWIPEV in Ny-Ålesund (78.9°N, 11.9°E), located on the west coast of Svalbard, at the south side of Kongsfjorden (Fig. 1). The area is mountainous, featuring seasonal snow cover, a typical tundra system, and glaciers. In the period investigated the sea has remained ice-free throughout the year. The local boundary layer is known to be strongly affected by the orography (Kayser et al., 2017; Beine et al., 2001), and is often quite shallow with an average mixing layer height below 700 m (Dekhtyareva et al., 2018; Chang et al., 2017). Surface layer temperature inversions are common, especially in winter (Maturilli and Kayser, 2017). The mountains reach up to 800 m,

and strongly influence the wind around Ny-Ålesund. In the free troposphere westerly winds prevail. The wind conditions are described more in detail in Sect. 3.4. Clouds have been found to occur above Ny-Ålesund 60–80 % of the time (Nomokonova et al., 2019b; Maturilli and Ebell, 2018; Shupe et al., 2011). Clouds generally occur more frequently in summer and autumn and are less common in spring, although the inter-annual variability is large.

## 2.2 Measurements and data products

Most of the measurements and cloud and thermodynamic parameter retrievals utilized were described in detail by Nomokonova et al. (2019b), and references therein. Here, the most important aspects are summarized, together with additional data products used. A summary of the instrumentation, their specifications and derived parameters is given in Table 1.

### 2.2.1 Instrumentation

We employ a suite of remote sensing instruments: radar, microwave radiometer and ceilometer. Within the frame of the (AC)[3]-project the JOYRAD-94 cloud radar was installed at AWIPEV on June 2016. In July 2017 it was replaced by the MIRAC-A cloud radar, which operated until October 2018. Both instruments are frequency modulated continuous wave cloud radars measuring at 94 GHz, described in detail by Küchler et al. (2017). The main difference between the two radars is the size of the antenna, which for MIRAC-A is only half of that of the JOYRAD-94. The smaller antenna leads to a sensitivity loss of about 6 dB and an increase of the beam width from $0.53°$ to $0.85°$ (Mech et al., 2019). A Humidity and Temperature PROfiler (HATPRO) passive microwave radiometer (MWR) has been operated continuously at AWIPEV since 2011. The instrument has 14 channels in the K- and V-bands to retrieve liquid water path (LWP), integrated water vapor (IWV), and temperature and humidity profiles (Rose et al., 2005). In addition to the zenith pointing measurements, an elevation scan is performed every 15–20 min to obtain more accurate temperature measurement in the boundary layer (Crewell and Löhnert, 2007). Finally, the Vaisala CL51 ceilometer measures at 905 nm providing attenuated backscatter coefficient ($\beta$) profiles (Maturilli and Ebell, 2018).

To compliment the remote sensing observations, we make use of soundings and standard meteorological parameters measured at the surface. In Ny-Ålesund radiosondes are launched routinely every day at 11 UTC, and more often during campaigns (Maturilli and Kayser, 2017; Dahlke and Maturilli, 2017). From the surface measurements we utilized temperature, pressure as well as wind speed and direction data (technical details in Table 1). The instruments for surface meteorology are continuously maintained by the AWIPEV staff, and all data is quality controlled (Maturilli et al., 2013).

### 2.2.2 Cloudnet

The Cloudnet algorithm combines radar, radiometer, and ceilometer with thermodynamic profiles from a numerical weather prediction (NWP) model to provide best estimates of cloud properties (Illingworth et al., 2007). The observational data, described in the previous section, is homogenized to a common resolution of 30 s in time and 20 m in the vertical. In the Ny-Ålesund dataset, the Global Data Assimilation System 1 (GDAS1, more info at https://www.ready.noaa.gov/gdas1.php)

was used as the NWP model until the end of January 2017, after which it was replaced by the operational version of the ICON (ICOsahedral Non-hydrostatic) NWP model (Zängl et al., 2015).

In our work we rely on the target classification product (Hogan and O'Connor, 2004), that classifies objects detected in the atmosphere as aerosols, insects, or different types of hydrometeors (cloud droplets, drizzle, rain, ice, melting ice; see Fig. 2 for an example). Radar reflectivity ($Z_e$) and ceilometer $\beta$-profiles are used to detect the presence and boundaries of clouds. Cloud phase is distinguished on the basis of $Z_e$, $\beta$, temperature ($T$) and wet bulb temperature; in addition, Doppler velocity from radar is used to position the melting layer. No differentiation is made between ice in a cloud and precipitating ice. While

applying this widely accepted methodology, for our study there are two important limitations. Firstly, the detection of liquid within a MPC is based on $\beta$, such that if cloud top is not found within 300 m from the height where the ceilometer signal is extinguished, all cloudy bins above this height are classified as ice. Secondly, no method to distinguish super-cooled drizzle from ice is available yet (Hogan et al., 2001; Hogan and O'Connor, 2004).

### 2.2.3 Derived properties

The amount of liquid and ice in the cloud, and their ratio, is one of the most important properties of MPCs. In addition, humidity supply is a key requirement for cloud formation and continuation. Liquid water path (LWP) and integrated water vapor (IWV) were retrieved from the zenith-pointing observations of the MWR using statistical multi-variate linear regression (Löhnert and Crewell, 2003). Coefficients for the retrieval were based on sounding data; more details about the retrieval and corrections applied are given by Nomokonova et al. (2019b). Previous studies have found the accuracy of the method to be 20–25 $\mathrm{gm}^{-2}$

(Löhnert and Crewell, 2003).

  Ice water content (IWC) was calculated using the $Z_e$-$T$-relationship from Hogan et al. (2006), where temperature was taken from the same model as used for Cloudnet. The uncertainty of the retrieval is estimated to be -33–+50 % for temperatures above -20°C. Ice water path (IWP) for P-MPCs was calculated by integrating IWC from the surface to cloud top. Furthermore, the LWP was averaged to 30 s to match the temporal resolution of IWP.

In order to calculate the potential temperature ($\theta$) profile based on the temperature profile retrieved from the MWR elevation scans, an estimate of the pressure profile is required. For this we took the measured surface pressure, and used the barometric height formula to estimate pressure at each height. The resulting $\theta$-profiles were compared with the profiles from radiosondes in the period June 2016–October 2018 (not shown). A slight cold bias is present (< 0.4 K). The RMSE increases with altitude, but in the lowest 2.5 km the RMSE is still below 1.8 K. For cloud top temperature, the temperature retrieved from the MWR

elevation scan was linearly interpolated between the retrieval levels to cloud top height.

# 3 Methods

## 3.1 Identification of persistent low-level mixed-phase clouds

To identify P-MPCs, each profile was evaluated individually to detect low pure-liquid and liquid-topped mixed-phase cloud layers, after which the persistency of the liquid layer was considered. Using Cloudnet target classification, the first step was to identify different cloud layers in each profile. Here a cloud layer refers to a continuous (gaps of <4 height bins, corresponding to 80 m, were omitted) layer of cloud droplets and/or ice. Each layer in the profile was classified as ice-only, liquid-only, or mixed-phase. To distinguish between low stratiform and deep multi-layered mixed-phase clouds, only profiles with a single liquid layer and the liquid layer being close to cloud top were considered. In practise, the detected upper boundary of the liquid layer was required to be in the uppermost 20 % of the cloud layer. The requirement for liquid being exactly at cloud top was relaxed since the ceilometer signal cannot necessarily penetrate the entire depth of the liquid layer. These criteria (single liquid layer, liquid close to cloud top) were very effective in selecting the desired low-level mixed-phase cloud regime. However, some mid-level clouds also fulfilled the criteria, and therefore we limited cloud top height to be below 2.5 km. For the remaining profiles, that all contain either liquid-only or liquid-topped mixed-phase clouds, the persistence of the liquid layer was evaluated. We only included clouds where the liquid layer existed for a minimum of one hour, with gaps $\leq 5$ min. Since the focus of this study is on mixed-phase clouds, we further excluded clouds where no ice was detected. Note, that continuous presence of ice was not required, only the cloud liquid had to persist in time. The result is a data set with clouds below 2.5 km where liquid is located at cloud top and persists at least one hour, and at some point in time ice is associated with the liquid layer. Note, that time periods where another cloud layer is found above the P-MPC are not excluded. Figure 2 shows an example of the identified persistent MPC as well as another mixed-phase cloud. Despite the strict criteria, such clouds were present 23 % of the observational time.

In addition to identifying the time periods with P-MPCs present, the Cloudnet data was used to determine the base of the liquid layer and the cloud top height. A P-MPC case was defined as the time from the beginning to the end of the identified persistent liquid layer. Furthermore, we consider the layer from liquid base to cloud top as the cloud and everything below liquid base to be precipitation. This definition was chosen because the liquid base is well defined from the ceilometer observations. Considering the focus on a persistent liquid layer identified by vertically pointing measurements, the cases included implicitly require either very low wind speeds, or a larger cloud field being advected over the site. When another cloud is detected above the P-MPC, the possibility that it contains undetected liquid cannot be excluded, and in these cases the measured LWP cannot be unambiguously attributed to the liquid layer of the P-MPC. Hence, those time periods were flagged to be removed in any analysis of the cloud's liquid content. Unfortunately, we cannot make the assumption that upper cloud layers would not impact the liquid content of a P-MPC (Shupe et al., 2013). The presented LWP distributions are therefore only representative for single-layer cases. Furthermore, all columns with liquid precipitation or drizzle were excluded, leading to a loss of data mainly in the summer months. While this is somewhat unavoidable (e.g. when the MWR measurements suffer from a wet radome), it leads to the exclusion of rather warm precipitating P-MPCs from the analysis.

## 3.2 Detecting surface coupling

### 3.2.1 Defining coupling with radiosonde profiles

The thermodynamic coupling of the P-MPC to the surface was determined based on the $\theta$-profile. A quasi-constant profile was taken to indicate a well mixed layer, while an inversion denotes decoupling between different layers. For the sounding profiles, we simplified the methodology of Sotiropoulou et al. (2014). The cumulative mean of $\theta$ from the liquid layer base height downward is compared to $\theta$ at each level below the cloud. If this difference exceeds 0.5 K, the cloud is considered decoupled. Fig. 3 a and b show two example cases, one for a coupled and one for a decoupled cloud, respectively. Both profiles demonstrate a structure typical for stratiform Arctic MPC: a temperature inversion at cloud top, below which a well mixed layer is identifiable. In the case of the coupled P-MPC (Fig. 3a), the well-mixed layer extends to the surface. For the decoupled P-MPC (Fig. 3b) the well-mixed layer extends 200 m below the liquid layer base, below which several weaker temperature inversions and a generally stable stratification can be identified.

### 3.2.2 New continuous method

To continuously evaluate the coupling of the P-MPC to the surface, we developed a new method based on surface observations and the potential temperature profiles retrieved from the MWR, which are available more frequently, i.e. every 15–20 min, compared to radiosonde data (Sect. 2.2.3). At each time when a MWR $\theta$-profile was available, the cloud was classified as either coupled or decoupled based on a two step algorithm. First, the stability of the surface layer was evaluated using the measurements of the the meteorological station. The premise of this criteria is that if the surface layer is stably stratified, the cloud must be decoupled from the surface as there exists a stable layer between the surface and cloud base. The $\theta$-profile is used as a proxy for stability. If the gradient of the 30 min mean $\theta$ between 2 and 10 m was positive (e.g. an inversion was present between 2 and 10 m), the surface layer was considered stably stratified, and therefore the cloud decoupled. If this was not the case, the second criteria based on the MWR $\theta$-profile was used. We calculate the difference in potential temperature ($\Delta\theta$) between the surface level and at the height half way to the liquid base height. If $\Delta\theta$ is below the threshold of 0.5 K, the cloud at this instance was considered coupled, and otherwise decoupled. The reason for using the height equaling half of the liquid layer base height can be understood by comparing the $\theta$-profiles from sounding and MWR in Fig. 3 a and b. While the general shape of the profile can be retrieved from the MWR measurements, it is not possible to resolve sharp inversions or detailed structures of the profile. Yet temperature inversions are very common at the top of P-MPCs. The comparison of MWR profiles with all available soundings when a P-MPC was present (Fig. 3c) shows that the accuracy of the retrieved potential temperature is reduced in the vicinity of the liquid layer top and that the influence extends to below the liquid layer base. At 0.5*liquid base height the impact of cloud top inversion is smaller than at liquid base and the RMSE is below 1 K, which is why we chose this height to determine the stability of the subcloud layer. Note, that it should not be inferred that the method can only detect decoupling occurring in the lowest half of the subcloud layer. When decoupling occurs above the layer explicitly included, it is common that the lower half of the subcloud layer is at least partly stably stratified, as can also be seen in the example of Fig. 3b, prompting a correct decoupling classification.

As the final step, the individual profiles were considered together to define the degree of coupling of each observed P-MPC case. For each detected cloud event, the number of coupled and decoupled profiles were counted. If all profiles were decoupled, the P-MPC was considered fully decoupled. When more than 50 % of the profiles were found decoupled, the P-MPC was defined as predominantly decoupled. The rest were considered coupled.

### 3.2.3 Comparison of methods

The performance of the new method for estimating the coupling for each individual profile was evaluated using the soundings as a reference. We restricted the soundings to cases for which the cloud was present from the launch time until the sonde passed a height of 2.5 km (maximum cloud top height considered). Those soundings were compared to the MWR profile closest (but not more than 20 min away) to the radiosonde launch time. The sounding-based diagnosis found 31 % of the evaluated P-MPCs coupled and 69 % decoupled, compared to 18 % and 82 %, respectively, for the newly developed method for the corresponding clouds (Fig. 3d). This suggests a tendency in our method towards decoupling. However, the sounding profiles may miss very shallow surface based inversions. For 24 % of the profiles considered as coupled based on the radiosondes, the 2 and 10 m temperatures indicate a surface inversion. Classifying these clouds as decoupled instead changes the ratio of coupled and decoupled P-MPCs from the sounding data set to 23 % and 77 %, which is closer to that found with the new method. The main disadvantage of our method is that the temperature profiles retrieved from the MWR measurements do not provide a detailed profile, rather the general shape of the profile, and so the developed method occasionally fails. Furthermore, the 10 m layer considered for surface stability is rather shallow and intermittent coupling could occur regardless of the thermodynamic profile structure.

### 3.3 Circulation weather type

Since the local wind direction in the lower troposphere above Ny-Ålesund is heavily influenced by the orography (Maturilli and Kayser, 2017), the circulation weather type based on Jenkinson and Collison (1977) was applied in order to evaluate cloud properties in the context of the regional wind field. Using 850 hPa geopotential height and shear vorticity from ERA-Interim, the flow at each time (0, 6, 12, and 18 UTC) was classified as either W, NW, N, NE, E, SE, S, SW, cyclonic, or anticyclonic. 16 grid points centered around Ny-Ålesund were used, so that the area covered is approximately 300 km in meridional and 100 km in zonal direction (77.5°–80.5° N, 9.75°–14.25° E, see Fig. 1a). The approach aids in assessing whether the observed clouds were advected to the site from the open sea or over the island, and the proximity of high and low pressure systems.

### 3.4 Local wind conditions

The channeling of the free-tropospheric wind along the fjord axis is a typical feature of an Arctic fjord (Svendsen et al., 2002; Esau and Repina, 2012, and references therein). Previous work has found the feature prominent also at Kongsfjorden (Maturilli and Kayser, 2017). It is well documented that despite the dominating westerly free-tropospheric wind direction, in Kongsfjorden the near surface wind tends to blow southeasterly along the fjord axis (Maturilli and Kayser, 2017; Beine et al.,

2001; Jocher et al., 2012). This is usually attributed to katabatic forcing of the Kongsvegen glacier about 15 km east-southeast from Ny-Ålesund (Fig. 1), although Esau and Repina (2012) argued that for typical synoptic conditions the land-sea breeze circulation would be the dominant driver. The secondary mode in surface wind is from northwest, from the sea towards the island's interior. According to Jocher et al. (2012) the northwesterly surface winds are associated to cold air advection that relate to passing low-pressure systems. Beine et al. (2001) find this wind direction to be pronounced in June and July, which they associate with sea breeze and the melting of sea ice. In addition, at Ny-Ålesund weak southwesterly surface winds are observed, caused by katabatic flow from the Zeppelin mountain range and the Broggerbreen glacier south of Ny-Ålesund (Jocher et al., 2012; Beine et al., 2001) under specific synoptic conditions (Jocher et al., 2012; Argentini et al., 2003). The local wind conditions impact the stratification of the local boundary layer (Argentini et al., 2003; Svendsen et al., 2002). Argentini et al. show that during the ARTIST campaign (15 March – 16 April 1998 at Ny-Ålesund) stable conditions were mainly observed with southeast wind and hardly ever with northwest wind. Unstable conditions occurred from 90°to 270°and under light wind conditions. Furthermore, large wind shear was observed to generate turbulence and lead to neutral stratification. This brief summary of previous studies demonstrates the complexities of the local wind conditions present at the AWIPEV station.

We cannot properly describe the circulation in Kongsfjorden from our point measurements at Ny-Ålesund or evaluate the drivers behind the local wind, nor are these processes within the scope of our study. However, it is possible that certain wind patterns are associated with phenomena (shear induced turbulence, drainage flows from mountains and glaciers transporting cold air into the sub-cloud layer) that modify the P-MPC studied. To evaluate whether the local wind patterns modify the P-MPC, we identified the main modes in the 10 m wind direction and combined them with the circulation weather type to create a proxy for different wind regimes. As expected, three modes can be identified in the surface wind (Fig. 4a). The dominating wind direction (85°–165°) corresponds to the direction out of the fjord to the open sea. Less pronounced but clearly identifiable are the two other modes that indicate flow from the sea into the fjord (270°–345°) and the katabatic flow from the glaciers south of the station (200°–270°). Wind speed above 12 ms$^{-1}$ was only observed between 90°and 120°. Seasonal wind roses are provided in Appendix A2. The frequency with which each surface wind mode was associated with the different weather types during the cloud observation period (June 2016 – October 2018) is illustrated in Fig. 4b. For most circulation weather types, the southeasterly surface wind dominated and the northwesterly was rare. An exception were the weather types N and NW, for which the northwesterly direction was most common.

To illustrate how the weather type and surface wind direction modes correspond to different wind profiles, the average wind direction profiles based on radiosonde data from June 2016 to October 2018 are shown for weather type W (Fig. 4c). When the surface wind direction was northwesterly, the average direction changed only slightly from the 280° in the free troposphere to align with the fjord axis at about 310°. The largest variation in the lowest 200 m was exhibited by the southwesterly surface wind direction. The most common regime (surface wind from southeast) had an average profile with free-tropospheric wind from the west, turning south and all the way to the southeast (120°) in the lowest 300 m. Figure 4c illustrates why a combination of surface and free tropospheric wind direction is needed to isolate different patterns. Considering the moderate standard deviation in the wind direction profiles shown in Fig. 4c, it is reasonable to assume that each surface wind direction

mode together with the weather type, which describes the mean regional wind direction at 850 hPa, describes a certain wind pattern, and thus gives a first estimate of the wind conditions around Ny-Ålesund.

## 3.5 Statistical tools

To test the statistical significance of differences between two or more distributions, the Mood's median test to compare the medians in different populations was used (Sheskin, 2000). This test was chosen because it does not require normally distributed data and the compared samples can be of different sizes. The median of each population is compared to the median of the distribution including all data, and the Pearson's $\chi^2$ test is used to test the null hypothesis that medians from different populations are identical. To reject the null hypothesis thus leads to the conclusion that the different populations have different medians.

The data points in the time series of the variables tested (LWP, IWP, cloud top temperature, and cloud base height) are correlated with each other, and can not as such be used in the statistical test. We assume that each P-MPC case is independent of each other, and for cloud top temperature and cloud base height use the medians for each case for testing. LWP and IWP were found to vary more within each case, and therefore several data points from every P-MPC case were sampled. For this, we estimated the de-correlation time scale as the time where the auto-correlation function, computed for each P-MPC case individually, reaches zero. For the majority of P-MPC cases there were too many gaps in the data to reliably compute the auto-correlation function, and hence no de-correlation time scale could be estimated. From the values available, the median was calculated and then double the median was used as the de-correlation time scale $\Delta t_{dcr}$ for all cases. For testing, the data was sampled randomly, with a minimum gap of $\Delta t_{dcr}$ between the sampled data points.

## 4 Results and discussion

### 4.1 Occurrence of persistent MPC and other clouds

We first examine the frequency of occurrence of different types of clouds in the observation period of the cloud radar (10 June 2016–8 October 2018) considering the 30 s averaged columns of the Cloudnet product. A cloud was found above Ny-Ålesund 76 % of the time measurements were running. The month-to-month variation was considerable, varying from 40 % to over 90 % (Fig. 5). Averaging for all years, cloudiness was slightly higher in summer (June–August; 80 %) and autumn (September–October; 77 %), and lower in spring (March–May; 69 %) and winter (December–February; 74 %). Intra-annual variation is pronounced in autumn, when cloud occurrence frequency varied from 69 % to 84 %. MPCs (defined here as any profile where co-located cloud liquid and ice are found) were present 41 % of the time, with a somewhat higher frequency in autumn. Liquid-only clouds (profiles with cloud droplets without co-located ice) had an overall occurrence frequency of 14 % and a clear seasonal cycle with most liquid-only clouds occurring in summer, and hardly any in winter or spring. Thus, the radiatively important cloud liquid was more often found in mixed-phase clouds, although the contribution of liquid-only clouds was notable in summer. All of the presented figures are given relative to the amount of data available. The top panel of Fig. 5

shows the high data coverage obtained, implying that - with the exception of the first and last month - we can give a reliable estimate of the frequency of cloud occurrence within the detection limits of the instruments.

The persistent low-level mixed-phase clouds (P-MPC, see Sect. 3.1 for definition) cover 23 % of the data set, highlighting the relevance of this cloud regime. In total 1412 cases of P-MPC were identified. The 'all MPC' and the P-MPC occurrences in Fig. 5 are not directly comparable, since the first one refers to individual profiles and the latter is to a large extent defined by a temporally continuous liquid layer and also includes profiles without a mixed-phase layer detected. P-MPC were most common in summer (32 %) and occurred less often in winter (15 %) and spring (16 %), with autumn being the intermediate season (24 %). The P-MPC occurrence thus follows the seasonal cycle of cloud liquid occurrence (Nomokonova et al., 2019b).

For defining the persistence of the liquid layer some thresholds needed to be set, including how long gaps were allowed, and the minimum duration required. The choices made (5 min and 1 h) were motivated by the aim for a certain cloud regime, namely a stratiform mixed-phase cloud in the boundary layer. A sensitivity test allowing only 2 min gaps in the liquid layer showed the only major difference being in the occurrence frequency of P-MPCs, while the properties of the clouds or the seasonal cycle of P-MPC occurrence did not differ substantially.

## 4.2 P-MPC properties and regional wind direction

Figure 6 shows the occurence of each weather type (used to determine the regional free-tropospheric wind direction, see Sect. 3.3) in our period of study, and the fraction of those times when a P-MPC was identified. In general, NE, SE and NW were less common than the other wind directions. For a given weather type, the fraction of P-MPC occurrence varied considerably. Almost a third of the time when winds were from west (W), a P-MPC was found at Ny-Ålesund. Weather types S, SW, NW, and anticyclonic were also favourable for P-MPC. Based on an evaluation of sounding profiles, the most common free-tropospheric wind direction for weather type anticyclonic was west (not shown). On the other hand, winds from north and east (weather types N, NE, and E) were less often bringing P-MPCs to the site. The weather types which are most commonly associated with P-MPCs can be determined by combining the occurrence frequency of each weather type and its P-MPC fraction (Fig. 6). Consequently, P-MPC were most often associated with the weather types W, SW, and anticyclonic, which include almost half (48 %) of all profiles.

The distributions of liquid layer base height, LWP and IWP and their dependence on wind direction are presented in Fig. 7. The base of the liquid layer was usually between 540–1020 m above the surface, with mean and median liquid base height of 860 and 760 m, respectively. The typical P-MPC thus lies above the fjord at a height fairly close to the mountain tops. Fewer P-MPC were associated with weather types NE, E and SE, and with mean liquid base heights well above 1 km these were found at larger altitudes than most of the P-MPC. The mean LWP for P-MPCs was 35 $\mathrm{gm}^{-2}$ with a standard deviation of 45 $\mathrm{gm}^{-2}$. On average most liquid was found in the P-MPC in weather type SW (49 $\mathrm{gm}^{-2}$), and least in weather type NE (12 $\mathrm{gm}^{-2}$). However, the variability within each weather type was larger than the differences between the weather types. The IWP distributions are strongly skewed (Fig. 7c) towards low values. Zeros were ignored, but all non-zero values were included. For all P-MPCs, the mean and median IWP were 12 and 2.1 $\mathrm{gm}^{-2}$, respectively. Between the different weather types, the mean (median) varied from the 5.6 (1.1) $\mathrm{gm}^{-2}$ of weather type SE to the 17 (6.2) $\mathrm{gm}^{-2}$ of weather type NW. The weather types NW,

W and SW stand out in terms of high IWP, and have a mean IWP of 16 $\mathrm{gm^{-2}}$. Overall, the westerly weather types (SW, W and NW) were associated with lower P-MPCs and with more liquid and ice (mean LWP 42 $\mathrm{gm^{-2}}$), while the easterly weather types (SE, E and NE) were less common, distinctly higher and connected to the lowest average LWP and IWP.

Large scale advection and air mass properties are known to influence MPC properties (Mioche et al., 2017; Qiu et al., 2018, amongst others). Previous studies suggest that at Svalbard northerly flow is often associated with cold air masses originating from the central Arctic, and that southerly flows bring warmer and more humid air from lower latitudes (Dahlke and Maturilli, 2017; Knudsen et al., 2018; Kim et al., 2017; Mioche et al., 2017). Furthermore, the open sea west of the Svalbard archipelago might act as a local source of humidity and heat. Here we use temperature at 1.5 $\mathrm{km}$ (corresponding to the 850 $\mathrm{hPa}$ level) and integrated water vapor (IWV) from the MWR to represent the atmospheric temperature and humidity conditions under which the P-MPC were occurring. In agreement with previous studies, Fig. 8 shows that the highest average IWV and warmest temperatures were associated with southerly winds, while the lowest average IWV and coldest temperatures with northerly winds. The domain considered for the weather type (Fig. 1a) is too small to describe large scale advection or air mass origin, but Fig. 8 suggests the weather type is nonetheless a useful proxy for air mass properties. The average IWV and 1.5 $\mathrm{km}$ temperature can explain the first order variation in P-MPC occurrence and LWP between weather types. The south-southwesterly winds are warm and humid, and are associated with frequent occurrence of P-MPC with relatively high amounts of liquid, compared to the north-northeasterly winds, which are drier and colder, and are associated with less frequent P-MPC occurrence and lower LWP (Fig. 6, 7b, and 8a). Owing to the complexity of ice micro-physical processes, such direct relationship cannot be found between atmospheric humidity and temperature (Fig. 8) and IWP (Fig. 7c). On the other hand, as already noted above, Fig. 7 shows a clear contrast between the properties of easterly and westerly P-MPC. These differences cannot be explained by the IWV and 1.5 $\mathrm{km}$ temperature distributions, which are rather similar for weather types W and E. Hence, atmospheric temperature and humidity are important, but not the only relevant forcing for P-MPC at Ny-Ålesund.

The influence of the island and its orography clearly affects the height of the liquid layer (Fig. 7a). The median altitude of the P-MPC base with easterly winds (weather types NE, E and SE) was above the height of the mountain tops, suggesting that the clouds usually were advected to the site above the mountains rather than forming locally in the fjord. The P-MPC associated with easterly winds were also less frequent (Figures 6 and 7a). If we assume the majority of observed P-MPC being of advective nature, the low occurrence frequency with winds from east would imply less cloud formation over the island compared to over the sea, or dissipation of cloud fields while being advected over the island. Mioche et al. (2015) found less low (below 3 $\mathrm{km}$) MPC over land than over sea in the Svalbard region in spring and winter, while in summer and autumn the differences were small. Cesana et al. (2012) studied liquid containing clouds in the Arctic, and found less low (below 3.36 $\mathrm{km}$) liquid containing clouds above Svalbard than over the surrounding sea in all seasons. Although direct comparison is not possible due to inconsistencies in the observation techniques, cloud sampling and the considered area, the mentioned studies all indicate that the influence of the Svalbard archipelago is to decrease the amount of low liquid bearing clouds.

The combination of the effects of large scale advection and air mass properties, as well as the influence of the Svalbard archipelago, can provide an explanation for the dependence of the P-MPC properties on weather type presented in Fig. 6 and 7. Southwesterly and westerly free-tropospheric winds were associated with most P-MPC and the highest average LWP and

IWP, likely due to higher amounts of humidity available from lower latitudes. The southeasterly to northeasterly winds had the least P-MPC, and comprise the lowest average LWP and IWP, related to the drier air masses from north and less favourable conditions for cloud formation over the island. Other mechanisms can be considered to further explain the observed IWP variation. Ice formation could be enhanced in the cold temperatures for weather types N and NE (Fig. 8), whereas the higher IWP for weather types SW, W and NW might be related to larger amounts of super-cooled liquid available in the P-MPCs (Fig. 7b,c) or higher aerosol concentration in airmasses advected from lower latitudes.

### 4.3 Seasonality

The seasonal variation of the studied P-MPC properties and atmospheric conditions at Ny-Ålesund are presented in Fig. 9. In agreement with previous studies (Nomokonova et al., 2019b; Maturilli and Kayser, 2017), the highest average temperature and humidity are found in summer, and the lowest in winter and spring (Fig. 9a,b). The height of the P-MPC shows a clear seasonality, with lower liquid base height in summer and higher in winter (Fig. 9c). Zhao and Wang (2010) evaluated five years of low-level clouds (cloud base below 2 km) observed at Utqiaġvik (previously known as Barrow), Alaska, and also found a seasonality in cloud height with a minimum in summer. Furthermore, these results are in agreement with the seasonality in cloud height at Ny-Ålesund reported by Shupe et al. (2011). The IWP distributions show a clear seasonality, with low values in summer and autumn and a clear maxima in spring (Fig. 9e). The low IWP in summer and autumn (median 0.2 and 1.0 $\mathrm{gm}^{-2}$, respectively) can be attributed to relatively warm temperatures close to $0°\mathrm{C}$. The median IWP in spring (7.5 $\mathrm{gm}^{-2}$) is almost 2-fold of the median IWP in the winter (4.0 $\mathrm{gm}^{-2}$), which can hardly be attributed to the different temperature conditions (Fig. 9b). The higher IWV in spring compared to winter (Fig. 9a), however, can play a role. Furthermore, the high IWP in spring could be related to the generally higher aerosol loading in the Arctic atmosphere in the late winter and spring, a time period also known as the Arctic haze season (Quinn et al., 2007).

On the contrary, the LWP distributions show a minimal seasonality despite the seasonal variation of IWV and 1.5 $\mathrm{km}$ temperature related to the P-MPC (Fig. 9a,b,d). The highest (lowest) median LWP in summer and spring (winter) was 24 $\mathrm{gm}^{-2}$ (18 $\mathrm{gm}^{-2}$), and the seasonal mean values varied from 33 to 36 $\mathrm{gm}^{-2}$. Note, that this result does not imply a lack of seasonal variability in overall cloud LWP (see Fig. 5 in Nomokonova et al., 2019a), only in the specific cloud regime evaluated. One challenge of the algorithm to identify the P-MPC are thick liquid layers where Cloudnet only identifies the lowest parts of the layer as containing liquid. The problem was partly mitigated by relaxing the criteria for liquid presence at cloud top, nonetheless we find cases with a thick liquid layer that do not fulfill the criteria of liquid topped mixed-phase layer and the rest of the cloud gets cut off (see Fig. 2 at 12:00 on 30 May 2018). This artificially limits our data set to clouds where the liquid layer is thin enough, and there might be some clouds with more liquid that are not included in our analysis. Considering the LWP distributions were skewed towards lower values (Fig. 7b), these cases are likely to be a minority for the cloud regime considered. However, it is possible that the average LWP is somewhat underestimated. In addition, it could be that the cloud detection algorithm limits the considered cases to a specific LWP regime, which results to the lack of seasonality in the LWP of the P-MPC.

Since P-MPC properties (excluding LWP) as well as atmospheric temperature and humidity vary seasonally, a seasonal dependency in wind direction could explain the weather type dependent variations in P-MPC properties found in Sect. 4.2. To examine this possibility, Fig. 10 shows the proportion of P-MPC observations in each season for every weather type. The observation period of 2.5 years from June 2016 to October 2018 together with the seasonal variation in P-MPC occurrence (Fig. 5) lead to the uneven distribution of data between seasons. Overall, the summer months contribute most to the data set. However, there are no extensive differences found between the weather types. Most noteworthy is the high spring and low autumn occurrence of NW, which might contribute to the high IWP for this weather type (Fig. 7c). Furthermore, N and E were relatively more common in winter, N and SE more common in spring, and N less common in autumn. Given the lack of a distinct signal, we believe the seasonal variation in wind direction plays a minor role in the weather type dependent differences in P-MPC occurrence and properties described in the previous section.

We further compare properties of the P-MPC at Ny-Ålesund and their seasonal variation to observations of similar cloud regimes at other Arctic sites. Only studies that comprise at least one year of observations were considered. Shupe et al. (2006) evaluated MPCs observed at the one year long Surface Heat Budget of the Arctic Ocean (SHEBA) campaign, and found an annual average LWP and IWP of 61 $\mathrm{gm}^{-2}$ and 42 $\mathrm{gm}^{-2}$, respectively. Both IWP and LWP were found to have a maximum in late summer and autumn. The study did not explicitly focus on low-level clouds, but found that 90 % of the observed MPC had cloud base below 2 $\mathrm{km}$. De Boer et al. (2009) focused on single-layer mixed-phase stratus at Eureka, Canada, and reported seasonal mean LWP to vary between 10 and 50 $\mathrm{gm}^{-2}$. Zhao and Wang (2010) show monthly mean values for LWP at Utqiaġvik to vary from 10 to 100 $\mathrm{gm}^{-2}$, and for IWP from 10 to 25 $\mathrm{gm}^{-2}$. Similarly to SHEBA, at both Eureka and Utqiaġvik the maximum LWP was found in autumn. However, at Eureka as well as Utqiaġvik a maxima in the amount of ice in MPCs was found in spring as well as autumn. The differences in seasonal cycles of LWP and IWP at different sites could be due to different forcing conditions, in addition to the choice of the cloud regime that might also play a role. Sedlar et al. (2012) included all single-layer clouds below 3 $\mathrm{km}$ and found that most of the LWP distribution was within 0 to 100 $\mathrm{gm}^{-2}$, with slightly higher values in the data set from SHEBA than Utqiaġvik. The average figures are comparable to those observed for P-MPC at Ny-Ålesund, although the mean values in our study are at the lower end of the range reported at Utqiaġvik and SHEBA.

Finally, the seasonal variation of P-MPC occurrence is compared with previous studies in the Svalbard region. Shupe et al. (2011) as well as Maturilli and Ebell (2018) report most clouds in summer and early autumn above Ny-Ålesund, agreeing with our findings. On the contrary, Mioche et al. (2015) identified most low-level (below 3 $\mathrm{km}$) MPC in the Svalbard region in autumn and a minimum in occurrence in summer based on the synergy of the measurements from CLOUDSAT and CALIPSO. P-MPC commonly contain very low amounts of ice which might be below the sensitivity limit of the satellite observations explaining some of the disagreement. Furthermore, Mioche et al. (2015) were missing clouds below 500 $\mathrm{m}$ due to the blind-zone of CLOUDSAT, and since clouds generally are lower in summer this would lead to a higher fraction of missed clouds in this season. In any case, considering the large month to month variation in cloud occurrence (also shown by Shupe et al., 2011), different results when considering different time periods can be expected. Our time series might still not be long enough to give a precise estimate of the seasonal variation of cloud occurrence frequency.

## 4.4 Surface coupling

Figure 11a shows the fraction of observed P-MPC classified as coupled, predominantly decoupled and fully decoupled in each season. 63 % of all observed P-MPC cases were found fully decoupled, and only 15 % were coupled. The degree of coupling had a clear seasonal cycle, with decoupling being the dominant mode in autumn and winter, and most coupled P-MPCs occurring in summer. The observed seasonality in the surface coupling of P-MPC could be related to the overall higher lower-tropospheric stability in winter, which could limit the coupling of the cloud. Previous studies have found that the coupling of low Arctic MPCs depends on the proximity of the cloud to the surface since the cloud driven mixing layer is more likely to reach the surface if the cloud is low (Shupe et al., 2013; Brooks et al., 2017). Also in our data set the median cloud base height for decoupled P-MPCs (1010 m) is considerably larger than the median cloud base height of the coupled P-MPCs (620 m) (Fig. 11b). For P-MPC with liquid base heights of more than 1.5 km coupling to the surface was not observed. P-MPC were on average higher in winter and lower in summer (Fig. 9c), which could partly explain the seasonal variation in the frequency of surface coupling.

To evaluate the effect surface coupling has on cloud properties, we only considered P-MPC in weather types SW and W in order to limit the different factors in play. These clouds include the full range of coupling states, and cover one third of the data set (Fig. 7a). The coupled P-MPC had more liquid than the fully and predominantly decoupled P-MPC (Fig. 12a). The median LWP did not differ significantly between the predominantly and fully decoupled P-MPC (25 and 28 $\mathrm{gm}^{-2}$, respectively), while the median LWP for coupled cases was clearly larger (47 $\mathrm{gm}^{-2}$). Differences in IWP between the coupling states were small (Fig. 12b). The medians did not vary significantly (from 11 to 12 $\mathrm{gm}^{-2}$), but the larger IWP values (between 30 and 100 $\mathrm{gm}^{-2}$) were less likely for the coupled P-MPC. From the LWP and IWP distributions it follows that the total amount of condensed water (LWP+IWP) was higher for coupled than predominantly or fully decoupled P-MPC. This suggests either a source of humidity from the surface that is not available for the decoupled P-MPC, or a smaller sink.

Many ice micro-physical processes have a temperature dependency (Lamb and Verlinde, 2011), and the observed differences in LWP and IWP distributions between coupled and decoupled P-MPCs could be caused by different sampling across the temperature range. Observed cloud top temperatures ranged from -28 to +5°C, with most P-MPC occurring at the warm end of this range (Fig. 12c). As the persistent liquid layer is the defining feature of the P-MPC, it is not surprising that they occurred more often at warmer temperatures where liquid is generally more abundant. The coldest P-MPC (cloud top temperatures below -18°C) were always decoupled, and were occurring in winter and early spring. The cloud top temperature distributions were very similar for the coupled and predominantly decoupled P-MPC, suggesting that the differences in IWP and LWP distributions between these two groups can not be explained by a varying frequency of different temperature regimes. The cloud top temperature distribution of fully decoupled P-MPC differs from that of the predominantly decoupled and coupled clouds by having larger number of cold cloud tops and a smaller peak at the warm end of the distribution. Yet the IWP and LWP distributions do not differ substantially between fully and predominantly decoupled P-MPC. Although the observed differences in LWP and IWP between coupled and fully decoupled P-MPC could be caused by differences in temperature, we

cannot explain the differences between predominantly decoupled and coupled, or the similarity of the predominantly and fully decoupled P-MPC simply from the cloud top temperature distributions.

The analysis presented only included weather types SW and W. These weather types are amongst the weather types having largest average LWP and IWP. The variation in LWP and IWP between coupled and decoupled P-MPC for the other weather types would therefore be smaller in absolute numbers. Including all weather types, the medians for LWP for coupled, predominantly and fully decoupled were 34, 22 and 20 $gm^{-2}$, and the medians for IWP 7.5, 9.4 and 9.4 $gm^{-2}$, respectively. The outcome that coupled P-MPC had more liquid, and that differences in IWP were small, is the same. One needs to keep in mind that these numbers are dominated by the westerly weather types which cover the bulk of the data. It is possible that different relationships between cloud properties of coupled and decoupled clouds would be found for weather types which have distinctly different mean wind conditions. While we cannot conclude that the presented results hold for all situations occurring at Ny-Ålesund, they describe the most common conditions.

The comparison between the coupling detection from sounding and the new method based on MWR and surface observations implied that the new method is more inclined to consider a profile decoupled (Sect. 3.2.3). Yet, the similarity of the LWP and IWP distributions for predominantly and fully decoupled P-MPC suggest that these groups were very similar. Considering cloud properties, it does not seem that the predominantly decoupled would be mistakenly considered more decoupled than they are. It is possible that our method erroneously classifies weakly coupled P-MPC as predominantly decoupled, and that in these cases the interaction with the surface is limited and does not modify the cloud properties considerably leading to similar LWP and IWP distributions for these clouds and the actually decoupled P-MPC. Accordingly, we conclude that decoupling might be overestimated, but this does not have serious consequences on the results on cloud properties. Considering the different estimates (Fig. 3d and 11a), we can regard 63–82 % of the P-MPC decoupled, and 15–33 % coupled. Moreover, intermittent turbulence and the coupling it may lead to are rather challenging for our approach, as the thermodynamic profile takes time to adjust. However, the turbulent transport of heat can be assumed to be similar as the transport of any other scalar. If the turbulence that occurred was too short-lived to modify the temperature profile distinctly, it would also be unlikely to transport great amounts of water vapor or aerosols to the cloud layer.

Shupe et al. (2013), Sotiropoulou et al. (2014) and Brooks et al. (2017) have evaluated the coupling of low clouds during the ASCOS campaign (August–September 2008) using different methods and slightly different time periods, and observed decoupling from surface 75 %, 72 % and 76 % of the time, respectively. Their measurements were mostly of clouds above sea ice, and for a shorter time period. The results are therefore not directly comparable with the multi-year statistic presented here. Moreover, the mechanisms that lead to decoupling at ASCOS were likely different than at Ny-Ålesund. Like Shupe et al. (2013), but unlike Sotiropoulou et al. (2014), we found a difference in LWP between coupled and decoupled clouds (Fig. 12a). If we assume that the (sea) surface can provide a source of moisture for the P-MPC, coupling could add moisture to the cloud layer and lead to a higher total water path. Considering the small differences in IWP (Fig. 12b), it does not seem that the surface would be an important source for INP, or there are some other mechanisms that limit ice formation in coupled clouds where more liquid water is present. The observed seasonality in the surface coupling of P-MPC (Fig. 11a) could be related to the

overall higher lower-tropospheric stability in winter, which could limit the coupling of the cloud, as well as to the lower cloud base height in summer (Fig. 9c) that makes it easier for the cloud to couple to the surface due to its proximity.

## 4.5 Local wind patterns around Ny-Ålesund

The effects local winds have on the P-MPC were evaluated using the weather type together with the surface wind direction as a proxy for the wind conditions at Ny-Ålesund (Sect. 3.4). The most common wind situation for the P-MPC at Ny-Ålesund is a southeasterly surface wind underlying westerly/southwesterly upper winds (Fig. 4, 6). Hence, the wind turns from the surface upwards to the almost opposing direction by 1.5 km height (Fig. 4c). Directional wind shear is therefore commonplace for P-MPC at Ny-Ålesund (Fig. 4b), either in or below the cloud layer. The magnitude of the wind direction change varies with the free-tropospheric wind. The only exception are weather types N and NW, for which the most common surface wind is northwesterly, and the wind does not turn, or only turns slightly, with increasing altitude. A further consideration related to the surface wind direction is the history of the boundary layer. The air has experienced very different surface properties when moving from open sea to land with northwesterly surface wind or from mountainous, often snow and ice covered terrain, to a flat sea surface with southeasterly surface wind.

The influence of local winds on the P-MPC was found to be limited. Figure 13a shows the fraction of time with P-MPC occurring (similarly to Fig. 6) for each weather type and surface wind direction combination. Weather types cyclonic and anticyclonic are somewhat hard to interpret, as these are associated with varying free tropospheric wind directions above the site, and were therefore not included. The low number of cases with southwest and northwest surface wind limits the possibilities to compare different surface wind regimes for most of the weather types. For weather types SW and W the southwest surface wind was associated with higher frequency of cloud occurrence compared to the southeast surface wind. In contrast, for weather type N northwest surface wind had P-MPCs most often and southwest the least. Based on this analysis no overall tendency for certain surface wind direction, or the amount of directional shear between the surface and the free-tropospheric wind, to increase or decrease P-MPC occurrence was found.

Regarding P-MPC properties, no strong relationships with surface wind direction were identified. Only the main findings are summarized here and further details are provided in Appendix A1. Considering weather types N, W and SW, which have the most cases across different surface wind directions, no statistically significant differences were found in the median liquid base height or cloud top temperature. The northwest surface wind was associated with the highest median LWP, possibly due to higher level of humidity available over the open sea. The southwest surface wind was associated with a significantly higher IWP for weather types W and SW (median IWP 16 $\mathrm{gm}^{-2}$ and 18 $\mathrm{gm}^{-2}$, respectively). However, these variations in LWP and IWP were not found for all three weather types analyzed.

Local winds in Kongsfjorden were quite apparently connected to the coupling of the P-MPC (Fig. 13b). Coupling was most common with northwest surface wind (from the sea) and least common with the southeast surface wind (towards the sea), and the same behaviour was found for every season despite the seasonality of both surface wind direction and cloud coupling (see Appendix A2). For the P-MPC to be thermodynamically decoupled from the surface, a stably stratified layer needs to exist between the surface and the cloud base. Argentini et al. report a dependence of surface layer stratification on wind direction

(Argentini et al., 2003, Fig. 5). Stable conditions were most often found with southeast surface wind, for which only 8% of the P-MPC were considered coupled. On the other hand, stable conditions were rare with northwest surface wind, for which 37% of the P-MPC were coupled. The near surface wind from southwest and southeast is often related with flows from the glaciers (Jocher et al. 2012; Beine et al. 2001; Sect. 3.4) that bring cold air down to the valley in a shallow layer close to the surface. Such a cold surface layer is very efficient in decoupling the cloud and acts against the cloud driven turbulence that could otherwise couple the P-MPC to the surface. This effect might be stronger with southeast than southwest surface wind, since the katabatic winds from southwest are weaker (Fig. 4a). The differences in the coupling of the P-MPC with varying wind conditions can be explained by the differences in stratification of the lower boundary layer under different surface wind conditions. We conclude that the surface wind has the potential to modify the conditions in the boundary layer, which in turn can act to suppress coupling.

The local influence on coupling makes assessing the connection between coupling and cloud properties more challenging. The cloud might have been coupled to the surface while over see, and when it was advected into the Kongsfjorden-valley the local wind changed in the sub-cloud layer leading to decoupling. It is also difficult to evaluate coupling and local winds separately, because most coupled clouds were associated with northwest surface wind (Fig. 13b). Coupled P-MPC had higher LWP than decoupled (Fig. 12a), and P-MPC associated with northwest surface wind had higher LWP than those occurring with other surface wind directions (Fig. A1b). Perhaps the higher LWP is related to the combined effect of the two: more humidity is available from the open sea than over land and coupling is required for the water vapor to be transported from the surface to the cloud layer. There is a relationship between surface coupling and the local wind conditions at Ny-Ålesund, but to understand the impact of the combined effects on P-MPC properties would require further studies.

## 5   Conclusions

We presented 2.5 years of vertically resolved cloud observations carried out at the AWIPEV station at Ny-Ålesund. Methods to identify persistent low-level mixed-phase clouds (P-MPC), their coupling to the surface as well as the regional and local wind conditions were developed. We found P-MPC to occur 23 % of the time, most often in summer and least often in winter. The cloud base was typically 0.54–1.0 km high, LWP 6–52 $\mathrm{gm^{-2}}$, and IWP 0.2–12 $\mathrm{gm^{-2}}$. P-MPC were found to occur at higher altitudes in winter and lower altitudes in summer. LWP presented a lack of seasonal variation, possibly due to the selection of the cloud regime in this study. On the other hand, IWP had a clear seasonal dependence. IWP was low in the relatively warm months of summer and autumn, and had a clear maxima in spring. The frequency of occurrence was found to depend on free-tropospheric wind direction, and most P-MPC were associated with westerly winds. The height of the cloud was strongly influenced by orography. Less frequent P-MPC and with higher cloud base height were found with easterly winds compared to westerly winds, and these clouds had lower LWP and IWP. The most common surface wind direction in Kongsfjorden is from southeast, but this is typically underlying synoptic winds from westerly directions. Local winds were not found to impact the occurrence or the height of the P-MPCs, but for some free-troposheric wind directions the surface wind direction was related to variations in LWP and IWP. P-MPC were mostly decoupled (63–82 % of the time), and coupling occurred most often in

summer and for clouds close to the surface. Coupled P-MPC had a higher LWP than decoupled P-MPC, but no differences in IWP were found. Furthermore, the local wind patterns appeared to be related to surface coupling, specifically, the P-MPC with surface wind directions associated with glacier outflows were more commonly decoupled. The variation of median LWP between different wind direction at 850 hPa was larger than the variation found between different surface wind regimes or coupling states. On the other hand, IWP was found to vary with regional and local wind direction as well as season, but no dependency with coupling to the surface was found. We conclude that while the regional to large scale wind direction was important for P-MPC occurrence and their properties, also the local scale phenomena such as surface coupling and the local flow in the fjord had an influence.

Our results suggest that the P-MPC water properties can be influenced by the processes in the local boundary layer. The observed LWP values are in the range where the clouds are not yet fully opaque, and changes in LWP will have an impact on the radiative forcing of clouds at Ny-Ålesund (Ebell et al., 2019). For numerical models to correctly describe low-level MPCs' ice and liquid water content, and hence the radiative effect, the boundary layer dynamics need to be accurately described. In Ny-Ålesund, and in other Arctic fjords, this requires that local wind in the fjord is represented, and thus a description of the orography and key surface properties (temperature, snow cover etc.) needs to be accounted for in the model.

Long-term datasets are valuable for evaluating models since the evaluation can be carried out in a statistical manner instead of case-by-case basis. The dataset presented in this paper can be used for model comparison, to provide insight on model performance regarding low-level MPCs in the complex Arctic fjord environment. In addition, the results presented here provide background information that aid the interpretation of case studies underway from recent measurement campaigns (Wendisch et al., 2018). In this study, the effects of aerosols acting as ice nucleating particles or cloud condensation nuclei have not been evaluated. Also the cloud micro-physical processes taking place should be considered in more detailed. Further work is thus needed to understand the relationships between various processes controlling the properties and development of low-level MPCs at Ny-Ålesund.

*Data availability.* The Cloudnet data are available at the Cloudnet website (http://devcloudnet.fmi.fi/). The radiosonde data are available in PANGAEA (doi:10.1594/PANGAEA.845373, Maturilli and Kayser (2016) for 1993-2014; doi:10.1594/PANGAEA.875196, Maturilli and Kayser (2017) for 2015-16; search term 'project:label:AC3 ny-alesund radiosonde' afterwards). The meteorological surface observations are available in PANGAEA under search term 'Continuous meteorological observations at station Ny-Ålesund'. The MWR data is also available in PANGAEA (10.1594/PANGAEA.902183, Nomokonova et al. (2019)). The software used for the median test was the courtesy of Keine (2019). The cloud micro-physical dataset is currently under rewiev for PANGAEA (https://doi.pangaea.de/10.1594/PANGAEA.898556, Nomokonova and Ebell (2019). Topography data in Fig. 1 are povided by Amante and Eakins (2009) (a) and Norwegian Polar Institute (2014) (b).

**Appendix A: Details on the relationship between local wind conditions and P-MPC**

**A1  P-MPC properties**

The results of the analysis on P-MPC properties for different wind regimes is provided here, and some possible mechanism are contemplated. The cloud properties associated with different surface wind directions were compared separately for each weather type. Only weather types N, W and SW were considered (Fig. A1) in order to have a sufficient amount of data (at least 30 cases) in each group being compared (Fig. 13a). The median liquid base height did not differ significantly (on a 95 % confidence level) for any of the three weather types evaluated. The northwest surface wind was associated with the highest median LWP, however, for weather type SW the differences were not statistically significant. For weather type N the median LWP for northwest surface wind was 22 $\mathrm{gm}^{-2}$ compared to the the 12 and 7.8 $\mathrm{gm}^{-2}$ of southeast and southwest surface winds, respectively. Also for weather type W the northwest surface wind was associated with highest median LWP (39 $\mathrm{gm}^{-2}$), however the lowest median LWP was with southeast surface wind (18 $\mathrm{gm}^{-2}$). The median IWP varied insignificantly (from 7.8 to 9.7 $\mathrm{gm}^{-2}$) for weather type N. For weather type SW, the southwest surface wind had the highest median IWP at 18 $\mathrm{gm}^{-2}$, almost double of the median of southeast (10 $\mathrm{gm}^{-2}$) and northwest (9.1 $\mathrm{gm}^{-2}$) surface winds. Similarly for weather type W the median IWP for southwest was 16 $\mathrm{gm}^{-2}$, and only 11 and 9.6 $\mathrm{gm}^{-2}$ for southeast and northwest surface winds. Because of the temperature dependence of many micro-physical processes, it would be possible that the observed differences were a result of different temperature regimes dominating in the compared groups. However, no statistically significant difference in the cloud top temperature distributions was found (Fig. A1d).

Local conditions evidently modify the wind field in the fjord (Sect. 3.4), but whether this affects the P-MPC is not as easily determined. Although we find some differences in the P-MPC occurrence and properties with different local low-level wind patterns (Fig. 13 and A1), these could also be due to the large scale conditions related to different local circulation patterns. We here consider some phenomena that might be taking place. The near surface wind from southeast could hinder the low P-MPC residing over the sea from advecting into the fjord where the observations were taking place. This would lead to higher cloud base height for the southeast surface wind regime, or a lower frequency of occurrence, as the lowest P-MPC would be limited. For both weather types N and W the northwest surface wind had the lowest 25-percentiles of the liquid layer base height (lower edge of the boxes in Fig. A1a). Fig. 13a gives no indication that the southeast surface wind would have been related to an overall lower frequency of occurrence. Although the lowest P-MPC were more often associated with northwest surface wind, liquid base height below 400 m was also not that common for this wind regime. Hence, it seems that the southeast surface wind was not substantially preventing the P-MPC on the sea from advecting into the fjord. Considering Figures 4c and A1a together, the depth of the layer where wind is found to deviate strongest from the free-tropospheric wind direction is below the median P-MPC base height, and the 25th percentile is above the depth of the layer where on average the wind is in alignment with the surface wind direction. Hence, many of the P-MPC reside in a layer where the wind direction is changing with altitude, or just above it. The wind shear could induce turbulence which in turn could affect the properties of P-MPCs, and it might be influencing vertical fluxes of heat, moisture, and aerosols. These kind of processes could explain the differences found in IWP

and LWP between different wind regimes. However, to examine these processes would require a more sophisticated description of the local circulation and turbulence in the boundary layer than was used here.

## A2 Seasonality of surface wind direction and P-MPC coupling

Seasonality in near surface wind and the degree of P-MPC coupling with different surface wind directions are presented. Figure A2 shows the windrose for each season for 10 m wind in the studied period, June 2016–October 2018. Differences between the seasons are present in the relative importance of the three surface wind modes, in agreement with Beine et al. (2001) and Maturilli and Kayser (2017). The summer months stand out with more common northwesterly winds, which has previously been attributed to sea breeze (Beine et al., 2001). Subsequently, the other directions are less frequent. In autumn and winter the northwesterly winds almost completely disappear. The seasonal variation is likely due to the different degree at which the drivers (e.g. sea breeze circulation, katabatic flow, channeling of free-tropospheric wind along the fjord) act in different seasons.

The relationship between surface wind and P-MPC coupling is similar in all seasons except summer (Fig. A2e-h). In winter, autumn, and spring coupling with southeast surface wind was rare or non-existent. Coupling mostly occurred with northwest surface wind. The reasons follow those given in Sect. 4.5: the more (less) stable stratification of the lower boundary layer associated with the southwest (northwest) wind, probably related to the cold outflow from the glaciers that increase the stability of the sub-cloud layer promoting decoupling. In summer the situation is somewhat different from the other seasons. Southeast wind was still related to the fewest coupled P-MPC, but the differences between different wind directions were smaller. Furthermore, the coupling frequency with southwest wind was very similar to the northwest wind. The wind roses for each season (Fig. A2a-d) suggest a variation in boundary layer dynamics in summer, which could be contributing to the altered relationship between surface wind direction and the frequency of P-MPC coupling. Moreover, as discussed in Sect. 4.4, the overall lower stability in the boundary layer as well as the lower cloud base height in summer enhance surface coupling compared to the others seasons. Hence, local wind conditions seem to have less importance in summer, although the interaction with the local boundary layer is present in all seasons.

*Author contributions.* RG did the method development, statistical analysis, visualization of the results and prepared the manuscript with contributions from all co-authors. UL, SK, and MS contributed with conceptualization, research supervision, and discussions of the results. KE oversaw data management at University of Cologne, and advised in the analysis and selection of data sets. MM provided the long-term radiosonde dataset and insights in the local conditions at Ny-Ålesund.

*Competing interests.* M. D. Shupe is an editor for the special issue Arctic mixed-phase clouds as studied during the ACLOUD/PASCAL campaigns in the framework of (AC)[3].

*Acknowledgements.* We gratefully acknowledge the funding by the Deutsche Forschungsgemeinschaft (DFG, German Research Foundation) – Projektnummer 268020496 – TRR 172, within the Transregional Collaborative Research Center "ArctiC Amplification: Climate Relevant Atmospheric and SurfaCe Processes, and Feedback Mechanisms (AC)$^3$". Contributions by Stefan Kneifel were funded by the German Research Foundation (DFG) under grant KN 1112/2-1 as part of the Emmy-Noether Group "Optimal combination of Polarimetric and Triple Frequency radar techniques for Improving Microphysical process understanding of cold clouds (OPTIMIce)". Contributions by Matthew Shupe were funded by the U.S. National Oceanic and Atmospheric Administration's Earth System Research Laboratory. We wish to thank the Finnish Meteorological Institute, the Aerosol, Clouds, and Trace Gases Research Infrastructure (ACTRIS), and especially Ewan O'Conner for the Cloudnet algorithm. Furthermore, this work would have not been possible without the contribution of Tobias Marke by his assistance regarding Cloudnet and providing the circulation weather type, as well as Tatiana Nomokonova, who compiled the cloud microphysical dataset and assisted in the use of the MWR data. We further wish to thank Christoph Ritter and Maximilian Mahn for sharing ideas and insightful discussions, and Birte Kulla for providing Fig 1b. Last but not least, we are grateful for the AWIPEV station staff for technical support, maintenance and operation of the instruments.

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

**Table 1.** The instruments used, the most relevant specifications of each measurement, together with an overview of derived parameters. If the vertical ($\Delta z$) or temporal resolution ($\Delta t$) is changed from that measured by the instrument, the resolution used in the analysis is given in the last column.

| | Instrument | Temporal resolution | Vertical resolution | Parameters measured | Derived parameters |
|---|---|---|---|---|---|
| **JOYRAD-94** | RPG-FMWC94-SP | 2–3 s | 100–400 m: 4 m<br>400–1200 m: 5.3 m<br>1.2–3 km: 6.7 m | Radar reflectivity ($Z_e$), Doppler velocity ($V_m$) | Cloud presence, cloud boundaries (by Cloudnet; $\Delta z = 20$ m, $\Delta t = 30$ s) |
| **MIRAC-A** | | 2–3 s | 100–400 m: 3.2 m<br>400–1200 m: 7.5 m<br>1.2–3 km: 9.7 m | | Ice water content ($IWC$) $\Delta z = 20$ m, $\Delta t = 30$ s |
| **Microwave radiometer** | HATPRO | 1 s | - | Brightness temperatures at 22.24–31.40 GHz | Liquid water path ($LWP$) |
| | | 15–20 min | - | Brightness temperatures at 51.26–58 GHz | Potential temperature ($\theta$) -profiles, $\Delta z = 50$–250 m in the lowest 2.5 km |
| **Ceilometer** | Vaisala CL51 | 12–20 s | 10 m | Attenuated backscatter ($\beta$) at 905 nm | Cloud base, liquid presence (by Cloudnet; $\Delta z = 20$ m, $\Delta t = 30$ s) |
| **Surfcae meteorology** | Thies Clima PT100 | | Measurement at 2 and 10 m | Temperature | Potential temperature ($\theta$) |
| | Paroscientific, Inc. 6000-16B | 1 min | - | Pressure | |
| | Combined Wind Sensor Classic, Thies Clima | | Measurement at 10 m | Wind speed and direction | 30 min mean wind speed and direction |
| **Radiosonde** | RS92, RS41 | At least 1/day | 5–7 m | Temperature, Pressure | Potential temperature ($\theta$) |
| | | | | Wind direction | Wind direction |

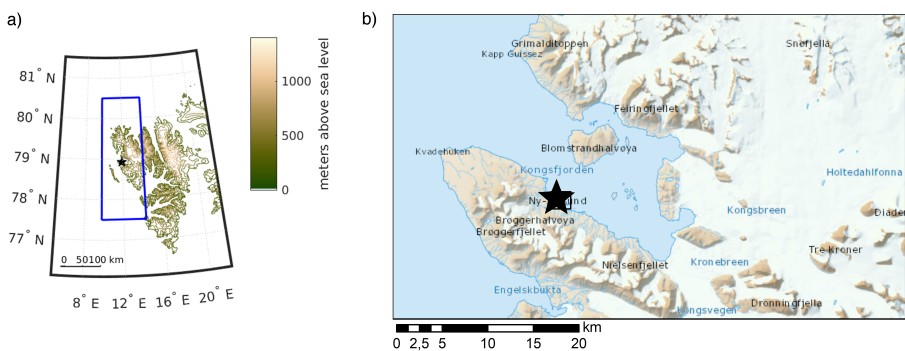

**Figure 1.** Topography map for Svalbard (a) and a detailed illustration of the Kongsfjorden area (b). The black star indicates the location of Ny-Ålesund, where the measurements are taken. The domain covered by the circulation weather type (see Sect. 3.3) is shown by the blue rectangle. Topography data by Amante and Eakins (2009) (a) and the Norwegian Polar Institute (2014) (b).

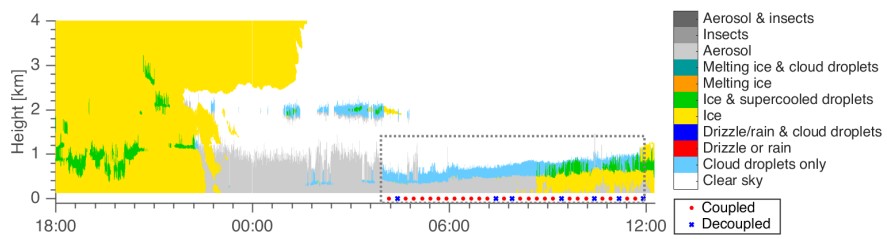

**Figure 2.** Example of the Cloudnet target classification product from 29 May, 18 UTC to 30 May 2018, 12 UTC. For the P-MPC, indicated by the gray dashed box, also the time series of coupling is shown. This case was classified as coupled.

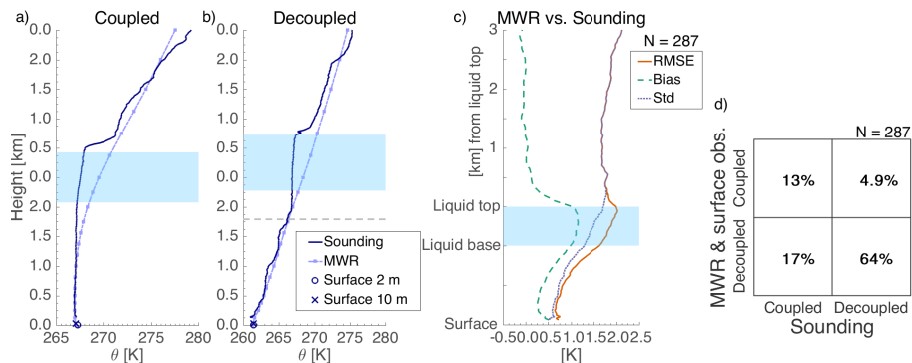

**Figure 3.** Examples of $\theta$-profiles from sounding and MWR, as well as surface observations: a coupled cloud on the 24 October 2017 11:55–12:05 UTC (a) and a decoupled cloud on the 1 February 2018 16:47–16:56 (b). The blue shaded area indicates the cloud layer, where cloud base and top are determined as the median values of the Cloudnet based cloud base and top for the duration of the sounding. The gray dashed line indicates the decoupling height defined from the sounding $\theta$-profile. Comparison of potential temperature profiles from sounding and retrieved from MWR measurements when P-MPC were present, with height normalized in respect to the liquid layer (c). Comparison of the diagnosed coupling with the new method based on MWR and surface observations and based on sounding profiles (d).

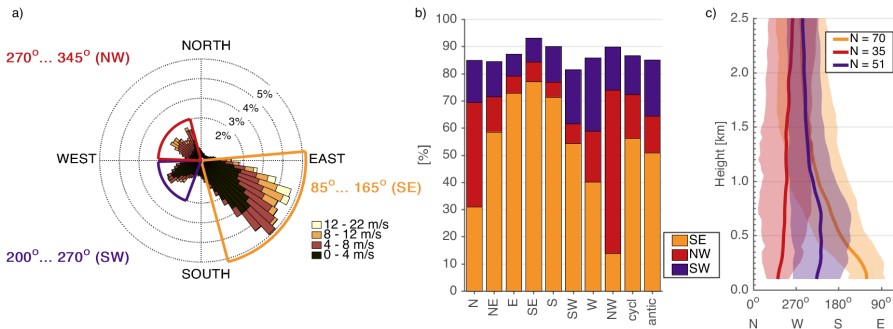

**Figure 4.** Wind rose for 30 min mean 10 m wind for the cloud observation period (June 2016–October 2018) with the three main modes identified (a), and the relative frequency of occurrence for each weather type (b). Wind direction profiles corresponding to each identified near surface wind direction mode for weather type W based on radiosoundings from June 2016 to October 2018 (c). The line shows the mean wind direction, and the shaded area the mean $\pm$ standard deviation at each height, estimated using the method by Yamartino (1984). Data points with wind speed below $0.5\,\mathrm{ms}^{-1}$ were omitted. N gives the number of soundings available for each mean profile.

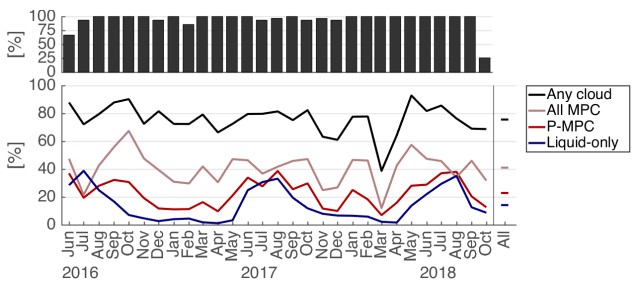

**Figure 5.** Monthly and total occurrence frequency of clouds in general and selected specific cloud types (see text for definitions) on bottom, coverage of Cloudnet data on top.

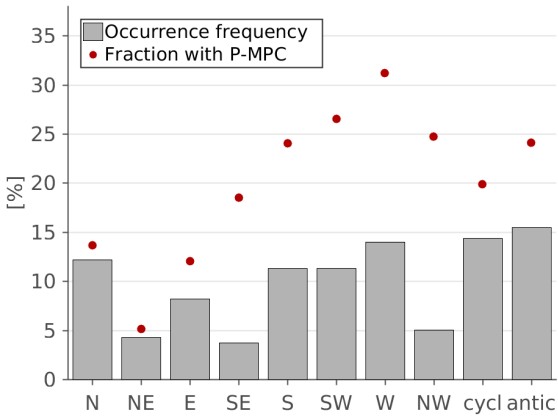

**Figure 6.** The frequency of occurrence of each weather type and the fraction with P-MPC presence.

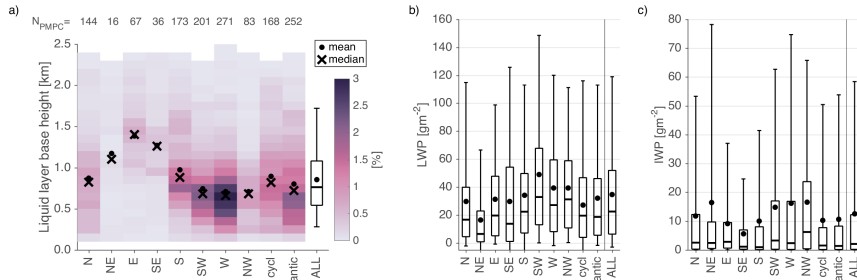

**Figure 7.** Height of the P-MPC liquid layer base (a), the LWP (b) and IWP (c) distributions for each weather type and all P-MPC. The number of P-MPC cases for each weather type is given in (a). The box shows the 25th, 50th and 75th percentile, the dot the mean, and the whiskers indicate the 5th and 95th percentile. The medians for different weather types were found to differ on a 95 % confident level.

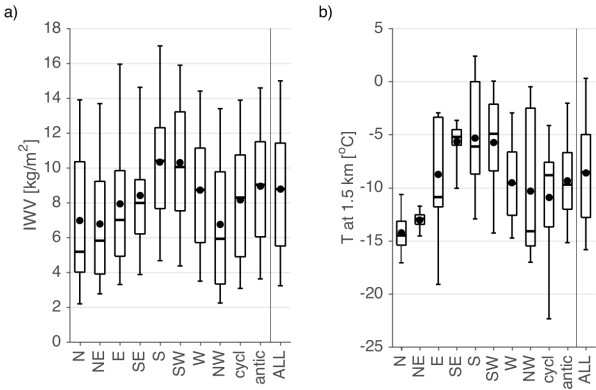

**Figure 8.** IWV (a) and 1.5 km temperature (b) for time periods with P-MPC present for each weather type. Boxes and whiskers as in Fig. 7. The medians were found to differ on a 95 % confidence level.

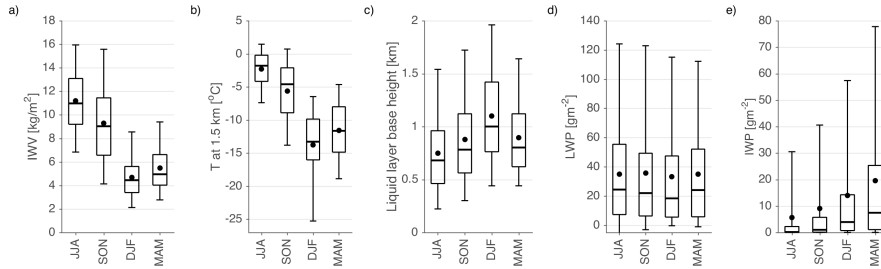

**Figure 9.** IWV (a), 1.5 km temperature (b), liquid base height (c), LWP (d) and IWP (e) distributions for each season. Only time periods with P-MPCs present are included. Boxes and whiskers as in Fig. 7; the medians were found to differ on a 95 % confident level.

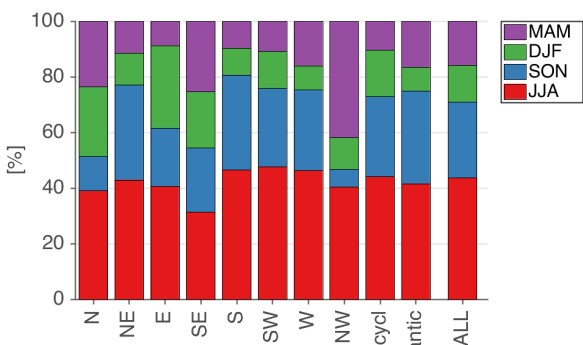

**Figure 10.** Distribution of seasons in the studied data set for each weather type. Only time periods when a P-MPC was present are included to evaluate the possible impact of wind direction seasonality on cloud properties and occurrence.

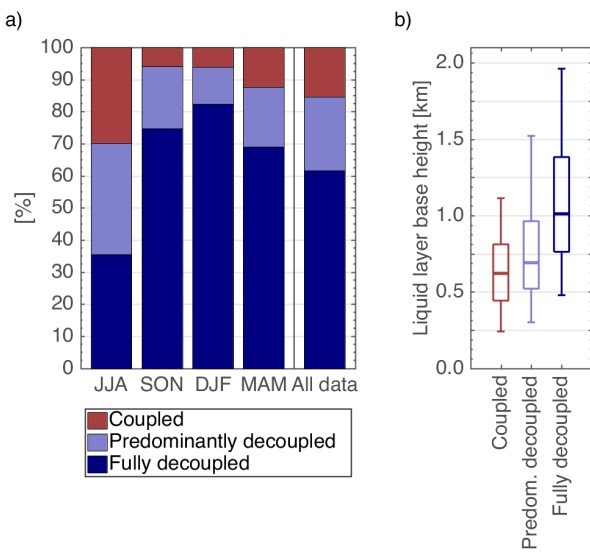

**Figure 11.** The fraction of P-MPC cases classified as coupled, predominantly decoupled and fully decoupled in each season and for the entire data set (a), and the distribution of the liquid layer base height in the coupling classes (b). Boxes and whiskers as in Fig. 7; the medians were found to differ on a 95 % confident level.

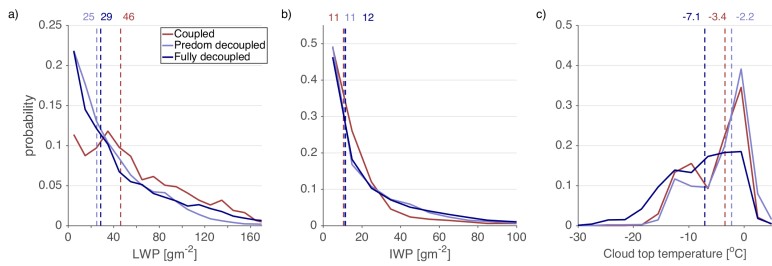

**Figure 12.** Comparison of LWP (a), IWP (b), and cloud top temperature (c) distributions between P-MPC in weather types W and SE with different degree of surface coupling. The dashed line and the numbers on top show the median value of each distribution. The bin size for LWP and IWP is $10 \text{ gm}^{-2}$ and for cloud top temperature $3°\text{C}$. The medians were found to differ on a 95 % confident level for LWP and cloud top temperature.

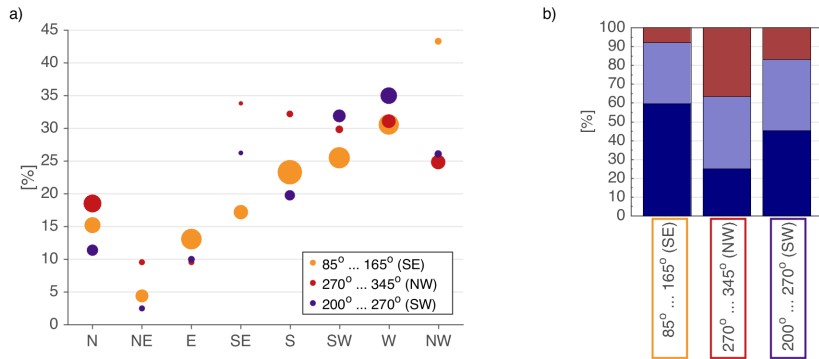

**Figure 13.** Fraction of time with P-MPC occurring for each surface wind direction and weather type regime (a). The size of the dots represent the amount of data available to compute the value. The fraction of P-MPC cases classified as coupled, predominantly decoupled and fully decoupled for each surface wind direction mode (b).

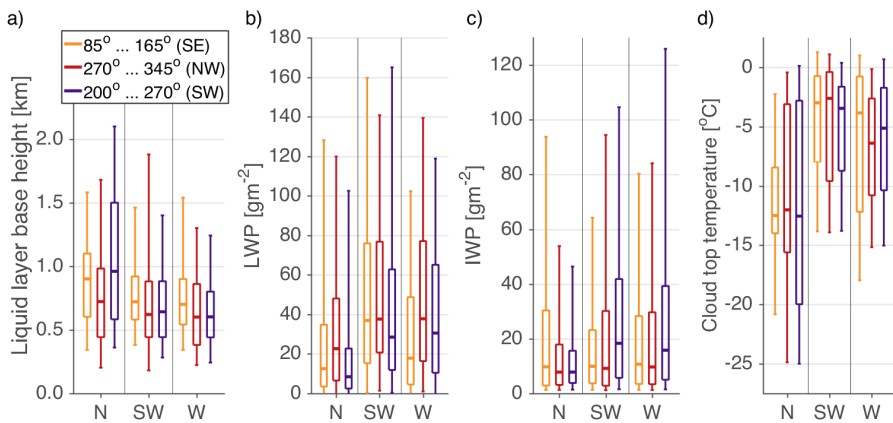

**Figure A1.** P-MPC liquid layer base height (a), LWP (b), IWP (c), and cloud top temperature (d) distributions for selected weather types and surface wind directions. Boxes and whiskers as in Fig. 7. The medians were found to differ (on a 95 % confidence level) in LWP for N and W, and IWP for SW and W.

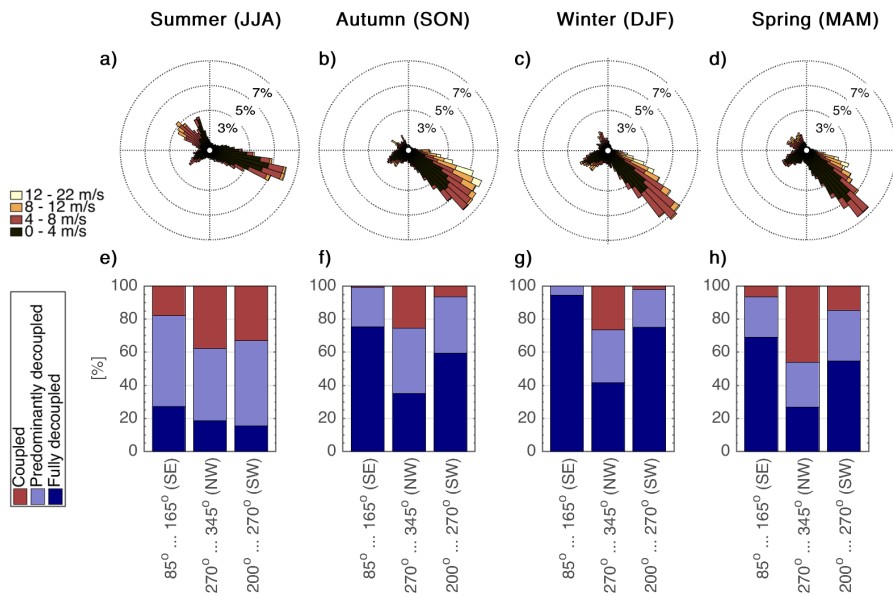

**Figure A2.** Wind rose for 30 min mean 10 m wind for each season (a-d) in the cloud observation period (June 2016–October 2018). The fraction of P-MPC cases classified as coupled, predominantly decoupled and fully decoupled for each surface wind direction mode in each season (e-f).