# Peer review of "Low-level mixed-phase clouds in a complex Arctic environment"

_Atmospheric Chemistry and Physics, 2019_

## Referee Comment (RC1) · Anonymous Referee #1 · 6 Aug 2019

Review of "Low-level mixed-phase clouds in a complex Arctic environment" by Gierens et al.

As far as I know this is a second paper reporting results from new 94 GHz cloud radar measurements at AWIPEV in Ny-Alesund after Nomokonova et al. (2019) who reported statistics of liquid, ice and mixed-phase clouds (MPC). This study reveals basic features on MPC using 2.5 year measurements, especially from a viewpoint of influences from the local meteorology. The authors showed that both liquid water path (LWP) and ice water path (IWP) tended to be higher under westerly wind conditions at 850 hPa. They also showed that local wind could affect coupling/decoupling of cloud layer with surface.

Because we still poorly understand behaviors of MPC in the Arctic and corresponding thermal structures of planetary boundary layer under different meteorological conditions, this paper is worth to be published in ACP. However, there are a few points which should be addressed before the publication, as described below.

Major comments:

(1) Moisture

This study describes three items, namely, properties of persistent MPC (P-MPC), local wind directions (surface and 850 hPa), and coupling/decoupling between the cloud layer and surface. Key parameters that connect these three items are temperature and moisture. Although some plots are shown for cloud top temperature, no data is presented for moisture. The authors may show integrated water vapor (IWV) obtained by MWR and atmospheric temperature to describe relationships among the three items.

For example, under the westerly conditions at 850 hPa (from open ocean), did the authors observe higher IWV and temperature as compared with those under easterly conditions (from island)? Did they observe higher IWV for coupled clouds as compared with those for decoupled clouds?

(2) Seasonality

In most of analyses, all data were used irrespective of month when the data was obtained. However, the authors may show the results from viewpoint of seasonality.

For example the authors may show seasonal variations of liquid layer base height, LWP, and IWP. (If they can also show cloud thickness and time duration of clouds (cloud persistence), it would be nice).

The authors may also show wind rose at 850 hPa in four seasons to show how the higher LWP under the westerly conditions at 850 hPa (Fig.8b) reflects the seasonal variations in wind direction.

Minor comments:

L.135: Explain what is "Ze".

L.135-137: The accuracy in LWP is described as 20-25 gm-2. Uncertainties in IWP is described to be -33 to +50%. Most of the differences in LWP and IWP for different wind directions presented in this study appeared to be within these uncertainties. What are the precision (uncertainties in relative values) in LWP and IWP estimations? Are the results presented in this study statistically significant?

L.194: What is the basis for this criteria? The authors may show some statistical results for vertical profiles of potential temperature in an appendix to show these criteria are reasonable.
Are all data with positive gradient in potential temperature discarded? No threshold value?

Figure 3: This figure is difficult to see. The authors may expand the figure to show the altitude range between 0 and 2km and time period between 03:00-12:00.

L.315 (Fig.8a): The authors may compare the cloud base height with lifting condensation level (LCL) calculated from surface measurements.

L.325: As far as I understood, the results described here are the most important one in this study. The authors may need to explain more on the physics behind. Higher LWP can be due to higher temperature (thermodynamic effect), moisture transport, lower stability (geometrically thicker clouds) and others. The authors may describe how these factors affect LWP under the westerly conditions, using IWV data and from viewpoint of seasonality.

Section 4.3: In my opinion, this section can be moved to appendix or deleted. Figure 9 clearly shows that 850 hPa wind direction is more important than that at surface. The

addition of surface wind analyses did not provide enough insights into the P-MPC.

L.487: What is the "observed differences"?

L. 542: According to Nomoknova et al., (ACP2019), mean value of IWP of MPC was 164 g m-2, while it was 12 gm-2. Why the values are so different?

L.544: Less and higher P-MPC -> Less frequent and higher cloud base height of P-MPC

L.554: The words "weather type" and "wind regime" are used in this study. Describe as "wind direction at 850 hPa" etc, such as in figure captions for Figure 4b.

---

## Referee Comment (RC2) · Anonymous Referee #2 · 8 Oct 2019

**Review for:**

"Low-level mixed-phase clouds in a complex Arctic environment"
*by Gierens et al.*

This study examines the influence of local topography and large-scale weather patterns on mixed-phase clouds (MPCs) observed at Ny Ålesund (Svalbard) for a 2.5 year period. They find that MPC occurrence is higher when westerly winds prevail; these clouds are also characterized by somewhat enhanced liquid and ice water contents. MPCs are also found more frequently decoupled from the surface; cloud-surface coupling (or decoupling) is attributed to local wind patterns.

The paper offers some useful insights regarding the behaviour and characteristics of Arctic MPCs. Some of the results could further be used for model evaluation. However, the comments below should be addressed before it is accepted for publication:

**Major Comments:**

(1) Section 4 is mainly a description of statistics. No attempt to physically interpret these results is made before section 5. I would recommend to the authors to discuss the underlying physical processes and how these are supported by the statistics during section 4. It is hard to remember all the details of this section when reading the recap in section 5.

(2) Seasonality is discussed in section 4.1 and regional wind patterns in section 4.2, but there is no attempt to investigate the links between these two factors. If a certain wind pattern dominates specific seasons, this should be considered when interpreting the results in section 4.2. The authors should present the frequency of the different wind patterns through the year.

(3) When using the MWR $\theta$-profile to assess decoupling, $\Delta\theta$ between the surface and the height half way to the liquid layer is estimated. But if decoupling occurs between this level and liquid base height, then the algorithm would classify a decoupled cloud as coupled. Do results vary significantly when using the gradient between the surface and the level exactly below the liquid layer as criterion? Please check the uncertainty in the applied method

(4) A main conclusion is that cloud-surface coupling is more frequent when wind comes from the sea and that it enhances cloud liquid. However for these conditions coupling occurs for somewhat less than 50% of the time. I suggest to the authors to investigate the meteorological conditions (e.g. large-scale moisture transport) between the decoupled and coupled cases when NW surface winds prevail. This might give indications of what drives coupling, which I don't think is the local wind pattern.

**Minor comments:**

Abstract: it is stated that westerly clouds had a higher mean liquid ($42\,\mathrm{g\,m^{-2}}$) and ice water path ($16\,\mathrm{g\,m^{-2}}$) compared to the overall mean of 35 and $12\,\mathrm{g\,m^{-2}}$, respectively. Is a $7\,\mathrm{g\,m^{-2}}$ difference in LWP important, given the large uncertainty in these retrievals? Moreover, I doubt that the impact on radiation is substantially different when changing cloud LWP by $7\,\mathrm{g\,m^{-2}}$. I don't think differences of this magnitude should be emphasized in the text.

Sections 2.3 and 3.3 offer a summary of the methods used to study the influence of the large-scale and the local wind patterns, repsectively. However the first method is included in Section 2 (Observations) and the second in Section 3. It would make more sense if section 2.3 becomes a subsection of Section 3, too.

Section 4.1: The percentages given in this section would be more meaningful if the actual number of PMPCs and all-PMC cases included in the analysis is stated. Is it same as in Figure 2?

L300-302: how different are the occurence statistics in the sensitivity test?

L481-484: here you combine Figure 5 and 10a to discuss how wind structure affects the PMPCs. However, Figure 5 corresponds to a much longer period than Figure 10a. For consistency, both wind and cloud measurements should correspond to the same time. Please check if utilizing fewer radiosondes results in very different wind structures. If this is the case, then you might consider removing/modifying the relevant discussion.

Figure 6: it would be better if the actual number of profiles is also included in the figure.

Figure 9: the size **of** the dots

---

## Author Comment (AC1) · 2 Jan 2020

We thank the two reviewers for their efforts and for the feedback that helped to improve the manuscript. A point-by-point response to all comments is given below (referee comments are highlighted in red). We have considered all comments made and a revised manuscript is submitted along this response. Figure and line numbers refer to the revised manuscript unless otherwise noted. Figures only appearing in this document are labelled with Roman numbers (e.g. I, II, III, …).

Sect. 3.2 *Detecting surface coupling* was extended to make the methodology and the reasoning behind it more understandable. Specifically, Fig. 2a in the original manuscript was replaced with two example cases. Following this the order of appearance of Figures 2 and 3 changed, so that Fig. 2 and 3 have swapped places in the revised manuscript. While the manuscript was in review, some additional microwave radiometer data become available and has been included in the analysis leading to a slight change in some values but not altering any of the conclusions of the paper. As advised by Referee #2, Sect. 2.3 *Circulation weather type* was moved under Sect. 3. In Sect. 3.4 *Local wind conditions* a paragraph was added to give more information on the local wind climate at the site based on the existing literature, and the description on the wind conditions based on sounding profiles was shortened. This also lead to excluding Fig. 5b and merging Fig. 5a with Fig. 4.

As suggested by Referee #2, the Results and Discussion (titled *A consolidated view of P-MPC at Ny-Ålesund* in the original manuscript) sections have been restructured so that the discussion is placed together with the presentation of the corresponding results. Sect. 4 is now titled "*Results and discussion*" with thematic subsections. Following the comments from both reviewers, a section on seasonality has been added, and the figure on the seasonality of cloud base height (Fig. 11c in the original manuscript) was moved from the section on coupling to this section. The section *Local wind patterns around Ny-Ålesund* has been moved to the end of the *Results and discussion* -section. The Results and discussion -section therefore now consists of five subsections: 4.1 *Occurrence of persistent MPC and other clouds*, 4.2 *P-MPC properties and regional wind direction*, 4.3 *Seasonality*, 4.4 *Surface coupling* and 4.5 *Local wind patterns around Ny-Ålesund*. The results regarding the connection of coupling and local winds have been moved from Sect. 4.4 to Sect. 4.5. Furthermore, the detailed results on the influences of local winds on cloud properties given in Sect. 4.5 *Local wind patterns around Ny-Ålesund* have been moved to the Appendix. Moreover, distributions of integrated water vapour and atmospheric temperature have been added in Sect. 4.2 following the comments from Referee #1.

The newly added Appendix A *Details on the relationship between local wind conditions and P-MPC* has two parts: A1 consists of the results and related discussion moved from Sect. 4.5.. Sect. A2 *Seasonality of surface wind direction and P-MPC coupling* presents additional seasonal figures for surface wind and occurrence of cloud coupling with different surface wind direction classes (same figures as Fig. 4a and 11b of the revised manuscript, respectively, but for each season).

A document detailing the changes made in the manuscript created by latexdiff is provided at the end of this document. In order to clearly communicate the changes in content, we first reorganised the text before running latexdiff. Hence, a block of text that was moved from one section to another does not appear as new text marked with blue.

**Reply to Referee #1**

(1) Moisture
This study describes three items, namely, properties of persistent MPC (P-MPC), local wind directions (surface and 850 hPa), and coupling/decoupling between the cloud layer and surface. Key parameters that connect these three items are temperature and moisture. Although some plots are shown for cloud top temperature, no data is presented for moisture. The authors may show integrated water vapor (IWV) obtained by MWR and atmospheric temperature to describe relationships among the three items.
For example, under the westerly conditions at 850 hPa (from open ocean), did the authors observe higher IWV and temperature as compared with those under easterly conditions (from island)? Did they observe higher IWV for coupled clouds as compared with those for decoupled clouds?

We thank the referee for this comment. While some description of the climatology of atmospheric moisture and temperature in the atmospheric column above Ny-Ålesund and the dependence on air mass origin is available in the literature (e.g. Dahlke and Maturilli, 2017), we agree that these important parameters ought to be described more in detail, especially in the context of the persistent mixed-phase clouds (P-MPC) that are the topic of this paper. As pointed out by the referee, this data is available from the micro-wave radiometer (MWR). We evaluated the distributions of integrated water vapour (IWV) and temperature at 1.5 km (corresponding to a typical height of the 850 hPa level) retrieved from the MWR, including only the time periods when P-MPCs were identified above Ny-Ålesund. The dependency of these parameters on regional wind direction (e.g. the weather type used in our study) and season have been included in Sections 4.2 and 4.3, respectively. To which extent the variations in atmospheric humidity and temperature can explain the observed variations in P-MPC occurrence and properties are shortly discussed in the corresponding sections.

To respond to the question on the relationship between P-MPC coupling and IWV, we have included Fig. I showing the IWV distributions for coupled, predominantly decoupled and fully decoupled P-MPC. Similarly to the cloud top temperatures presented (Fig. 12c), the coupled and predominantly decoupled occurred in rather similar humidity conditions, while the fully decoupled are associated with slightly drier atmospheric conditions. This is in agreement with the other results showing that the fully decoupled P-MPC have lower cloud top temperature, and occur most often in winter when IWV is lower. Analogous to the cloud top temperature, the IWV distributions are not able to explain the similarity of the IWP and LWP distributions of fully and predominantly decoupled P-MPC, or the differences between the coupled and predominantly decoupled P-MPC.

[Figure]

**Figure I.** Distributions of IWV for coupled, predominantly decoupled and fully decoupled P-MPC. The dashed line and the numbers on top show the median value of each distribution. The bin size is $3\,\mathrm{kgm^{-2}}$. The medians were found to differ on a 95 % confidence level.

*Changes in the manuscript:*

- Figure 8 was added to show the atmospheric humidity and temperature connections associated with the P-MPC and different weather types. The following text was added:

  "Here we use temperature at 1.5 km (corresponding to the 850~hPa level) and integrated water vapor (IWV) from the MWR to represent the atmospheric temperature and humidity conditions under which the P-MPC were occurring. In agreement with previous studies, Fig. 8 shows that the highest average IWV and warmest temperatures were associated with southerly winds, while the lowest

average IWV and coldest temperatures with northerly winds. The domain considered for the weather type (Fig. 1a) is too small to describe large scale advection or air mass origin, but Fig. 8 suggests the weather type is nonetheless a useful proxy for air mass properties. The average IWV and 1.5 km temperature can explain the first order variation in P-MPC occurrence and LWP between weather types. The south-southwesterly winds are warm and humid, and are associated with frequent occurrence of P-MPC with relatively high amounts of liquid, compared to the north-northeasterly winds, which are drier and colder, and are associated with less frequent P-MPC occurrence and lower LWP (Fig. 6, 7b, and 8a). Owing to the complexity of ice micro-physical processes, such direct relationship cannot be found between atmospheric humidity and temperature (Fig. 8) and IWP (Fig. 7c). On the other hand, as already noted above, Fig. 8 shows a clear contrast between the properties of easterly and westerly P-MPC. These differences cannot be explained by the IWV and 1.5 km temperature distributions, which are rather similar for weather types W and E. Hence, atmospheric temperature and humidity are important, but not the only relevant forcing for P-MPC at Ny-Ålesund." (L. 355-368)

[Figure]

**Figure 8.** IWV (a) and 1.5 km temperature (b) for time periods with P-MPC present for each weather type. The box shows the 25th, 50th and 75th percentile, the dot the mean, and the whiskers indicate the 5th and 95th percentile. The medians were found to differ on a 95 % confidence level.

- The Sect 4.2 is now summarized in the last paragraph as follows:

"The combination of the effects of large scale advection and air mass properties, as well as the influence of the Svalbard archipelago, can provide an explanation for the dependence of the P-MPC properties on weather type presented in Fig. 6 and 7. Southwesterly and westerly free-tropospheric winds were associated with most P-MPC and the highest average LWP and IWP, likely due to higher amounts of humidity available from lower latitudes. The southeasterly to northeasterly winds had the least P-MPC, and comprise the lowest average LWP and IWP, related to the drier air masses from north and less favourable conditions for cloud formation over the island. Other mechanisms can be considered to further explain the observed IWP variation. Ice formation could be enhanced in the cold temperatures for weather types N and NE (Fig. 8), whereas the higher IWP for weather types SW, W and NW might be related to larger amounts of super-cooled liquid available in the P-MPCs (Fig. 7b,c) or higher aerosol concentration in airmasses advected from lower latitudes." (L. 380-388)

- The source of the IWV was added in Sect. 2.2.3 (new text in bold):

"**In addition, humidity supply is a key requirement for cloud formation and continuation.** Liquid water path (LWP) **and integrated water vapor (IWV) were** retrieved from […]" (L. 135-137)

(2) Seasonality
In most of analyses, all data were used irrespective of month when the data was obtained. However, the authors may show the results from viewpoint of seasonality.
For example the authors may show seasonal variations of liquid layer base height, LWP, and IWP. (If they can also show cloud thickness and time duration of clouds (cloud persistence), it would be nice).
The authors may also show wind rose at 850 hPa in four seasons to show how the higher LWP under the westerly conditions at 850 hPa (Fig. 8b) reflects the seasonal variations in wind direction.

We understand this comment to consists of three main points, namely: 1) a suggestion to consider the seasonality of the presented properties, and to show the seasonal variation of cloud base height, LWP and IWP, 2) a proposal to also consider the seasonal variation of additional cloud properties, i.e. geometrical thickness and duration of the P-MPC, and 3) the advice to demonstrate how the  seasonality in 850 hPa wind direction is related to the relationship between LWP and wind direction presented in Sect 4.2. We respond to these three remarks one by one.

The manuscript only considered the seasonality of cloud occurrence and surface coupling of P-MPC. Additionally, the seasonal variation of cloud base height was shown because this was relevant to correctly interpret the variation of cloud base height of coupled and decoupled clouds, as coupling was found to be seasonally dependent. As noted by the referee, seasonal variation of P-MPC properties in themselves were not thematized. To correct for this shortcoming, we have added a section (Sect. 4.3) describing the seasonality of P-MPC properties, as well as atmospheric humidity and moisture due to their role as key air mass properties related to the P-MPCs. We have moved relevant parts of the originally submitted manuscript to this section (specifically: the seasonal variation of cloud base height, comparison with results from other sites as well as previous studies from Svalbard region), as well as added a description of the seasonality of LWP, IWP, IWV and 1.5 km temperature (Fig 9, L. 390-415).

For the second aspect, considering the seasonal variation of P-MPC geometrical thickness and duration, these variables are presented in Fig. II. The depth of the P-MPC, defined here as the distance from liquid cloud base to cloud top (the last bin detected by either radar or ceilometer), did not show notable seasonal variation. The mean P-MPC depth varied from 360 to 390 m. Due to the lack of seasonality we don't find this result interesting enough to include in the manuscript. Secondly, the duration of each P-MPC case (L. 172-173: "A P-MPC case was defined as the time from the beginning to the end of the identified persistent liquid layer.") shows seasonal variation. On average, the cases are shorter in winter and longer in duration in summer. However, individual very persistent cases are found in all seasons, and the longest persisting case (63 h) occurred in winter. Since our P-MPC identification algorithm is heavily relying on the persistence of a liquid layer, the thresholds set in the algorithm have a direct influence on the P-MPC duration. Let us consider an example cloud that has a liquid layer persisting for 3 h with a 4 min gap half way. If we only allow gaps up to 3 min this cloud would be considered as 2 cases with a duration of 1 h 28 min each. If instead a 5 min gap is allowed, only one case with a duration of 3 h is identified. This example illustrates how cloud persistence is build in our P-MPC detection algorithm. Although we found a similar seasonal cycle in our sensitivity tests for the persistence criteria, we are cautious to present quantitative results on P-MPC duration and this result was not included in the revised manuscript.

[Figure]

[Figure]

**Figure II.** Seasonal variation of P-MPC thickness (left) and duration (right).

Lastly, the referee pointed out that it would be worth to consider the seasonal variation in the 850 hPa wind direction, and whether this could explain some of the relationships between wind direction and P-MPC properties (specifically LWP) identified in Sect. 4.2. We have considered this possibility and found no seasonal variation in the weather type (used to describe the regional wind field at 850 hPa) that could explain the observed differences in cloud occurrence and properties between different weather types. To allow others to make the same conclusion, we have added a figure presenting the seasonal occurrence of all weather types (Fig. 10) in the manuscript.

*Changes in the manuscript:*

Section 4.3 *Seasonality* (L. 389-450) was added to describe the seasonal variability of key parameters. To avoid copying the entire section here, we only summarize the content of the section and provide the figures and new results here. For complete Sect. 4.3 we kindly ask to consult the attached manuscript.

- Sect. 4.3 includes the description of seasonality of the cloud base height, which was previously presented together with P-MPC coupling and the relationship between coupling and cloud base height (L. 392-393). In addition, Fig. 9 includes the seasonal variation of IWV, 1.5 km temperature, LWP and IWP, together with the following texts:

  "The seasonal variation of the studied P-MPC properties and atmospheric conditions at Ny-Ålesund are presented in Fig. 9. In agreement with previous studies (Nomokonova et al., 2019b; Maturilli and Kayser, 2017), the highest average temperature and humidity are found in summer, and the lowest in winter and spring (Fig. 9a,b)." (L. 390-392)

  "The IWP distributions show a clear seasonality, with low values in summer and autumn and a clear maxima in spring (Fig. 9e). The low IWP in summer and autumn (median 0.2 and 1.0 $gm^{-2}$, respectively) can be attributed to relatively warm temperatures close to 0°C. The median IWP in spring (7.5 $gm^{-2}$) is almost 2-fold of the median IWP in the winter (4.0 $gm^{-2}$), which can hardly be attributed to the different temperature conditions (Fig. 9b). The higher IWV in spring compared to winter (Fig. 9a), however, can play a role. Furthermore, the high IWP in spring could be related to the generally higher aerosol loading in the Arctic atmosphere in the late winter and spring, a time period also known as the Arctic haze season (Quinn et al., 2007).
  On the contrary, the LWP distributions show a minimal seasonality despite the seasonal variation of IWV and 1.5 km temperature related to the P-MPC (Fig. 9a,b,d). The highest (lowest) median LWP in summer and spring (winter) was 24 $gm^{-2}$ (18 $gm^{-2}$), and the seasonal mean values varied from 33

to 36 gm$^{-2}$. Note, that this result does not imply a lack of seasonal variability in overall cloud LWP (see Fig. 5 in Nomokonova et al., 2019a), only in the specific cloud regime evaluated." (L. 396-406)

[Figure]

**Figure 9.** IWV (a), 1.5 km temperature (b), liquid base height (c), LWP (d) and IWP (e) distributions for each season. Only time periods with P-MPCs present are included. Boxes and whiskers as in Fig. 7; the medians were found to differ on a 95 % confident level.

- The consideration of the P-MPC detection algorithms limitation regarding to thick liquid layers was moved to this section from 4.2 P-MPC properties and regional wind direction, and a sentence commenting on the possibility of the cloud identified cloud regime leading to the lack of seasonality in LWP was added:

"In addition, it could be that the cloud detection algorithm limits the considered cases to a specific LWP regime, which results to the lack of seasonality in the LWP of the P-MPC." (L.412-414)

- To clearly demonstrate that the dependency of P-MPC on weather types presented in Sect. 4.2 is not a result in seasonal variation of P-MPC occurrence and properties combined with a seasonality in 850 hPa wind direction, Fig. 10 is included with the following text:

"Since P-MPC properties (excluding LWP) as well as atmospheric temperature and humidity vary seasonally, a seasonal dependency in wind direction could explain the weather type dependent

[Figure]

**Figure 10.** Distribution of seasons in the studied data set for each weather type. Only time periods when a P-MPC was present are included to evaluate the possible impact of wind direction seasonality on cloud properties and occurrence.

variations in P-MPC properties found in Sect. 4.2. To examine this possibility, Fig. 10 shows the proportion of P-MPC observations in each season for every weather type. The observation period of 2.5 years from June 2016 to October 2018 together with the seasonal variation in P-MPC occurrence (Fig. 5) lead to the uneven distribution of data between seasons. Overall, the summer months contribute most to the data set. However, there are no extensive differences found between the weather types. Most noteworthy is the high spring and low autumn occurrence of NW, which might contribute to the high IWP for this weather type (Fig. 7c). Furthermore, N and E were relatively more common in winter, N and SE more common in spring, and N less common in autumn. Given

the lack of clear trends, we believe the seasonal variation in wind direction plays a minor role in the weather type dependent differences in P-MPC occurrence and properties described in the previous section." (L. 416-425)

- The comparison of the results with those from other sites and previous studies in the Svalbard area were included in this section (L. 515-537 in the original manuscript), and extended to consider seasonality.

- In the Abstract the following sentence was added:

"Seasonal variation of the liquid water path was found to be minimal, although the occurrence of persistent MPCs, their height and ice water path all showed notable seasonal dependency." (L. 11-12)

- In Sect. 5 *Conclusions* the following was added:

"P-MPC were found to be higher in winter and lower in summer. LWP presented a lack of seasonal variation, possibly due to the selection of the cloud regime in this study. On the other hand, IWP had a clear seasonal dependence. IWP was low in the relatively warm months of summer and autumn, and had a clear maxima in spring" (L.574-577)

L.135: Explain what is "Ze".

Ze is defined when first mentioned in Section 2.2.2, on p. 5 L. 127.

L.135-137: The accuracy in LWP is described as 20-25 g m$^{-2}$. Uncertainties in IWP is described to be -33 to +50 %. Most of the differences in LWP and IWP for different wind directions presented in this study appeared to be within these uncertainties. What are the precision (uncertainties in relative values) in LWP and IWP estimations? Are the results presented in this study statistically significant?

A relatively high level of uncertainty is an unfortunate aspect of the retrievals available. We compared all distributions presented following the methods described in Sect. 3.5 *Statistical tools*, and all figure captions include the information which medians were found to differ on a statistically significant level. All results described in the text are statistically significant, unless described as "not significant", for example "The median LWP did not differ significantly between the predominantly and fully decoupled P-MPC" (L. 465-466) and "The medians did not vary significantly" (L. 468). The word *significant* is reserved to only describe statistical significance. We have now double checked the use of the word in the manuscript, and the one omission to this practice has been corrected.

*Changes in the manuscript:*
L. 313: significant → notable

L.194: What is the basis for this criteria? The authors may show some statistical results for vertical profiles of potential temperature in an appendix to show these criteria are reasonable.
Are all data with positive gradient in potential temperature discarded? No threshold value?

The referee is questioning the basis of the criteria for the stability of the surface layer, namely using the gradient of the 30 min mean potential temperature between 2 and 10 m. Using the potential temperature profile for estimating stratification in the surface layer does not differ from the principle from using the potential temperature profile for the entire subcloud layer. A positive gradient is used in both cases as an indicator for a stable stratified layer. The threshold value is explicitly zero, above this value the layer is considered stable stratified (and with that the cloud decoupled from the surface). As a time series is available from the surface measurements, a 30 min mean was used to ignore the impact of small scale fluctuation caused by individual eddies. This averaging period is common for micro-meteorological studies in the boundary layer (Stull, 1988, p. 33). No data is discarded based on the surface layer stability criteria, only a classification for a decoupled cloud is made. If the cloud is not found decoupled based on the surface layer stability criteria, the second criteria based on MWR is used.

To reply to the question if the criteria whether the potential temperature gradient ($\Delta\theta$) is reasonable, we compared it with the the bulk-Richardson number ($Ri_B$)

$$Ri_B = \frac{g\,\Delta\theta\,\Delta z}{\theta\,(\Delta U)^2} \qquad (1)$$

for the 2 to 10 m layer using the surface meteorological data (Stull, 1988, p. 177). The Richardson number describes the ratio of buoyant and mechanical production of turbulent kinetic energy. A critical value $Ri_{crit}$ of 0.25 is used to define stable stratification. The relationship between the temperature gradient and $Ri_B$ is presented in Fig. III. The two approaches for determining surface layer stability agree 72 % of the time. Disagreement occurs when $\Delta\theta$ is positive, but $Ri_B$ indicates a turbulent layer due to high wind shear. Considering also the limitations related to $Ri_B$, we think the use of $\Delta\theta$ is justified. Furthermore, using $\Delta\theta$ in the surface layer is consistent with the second criteria used for coupling, which is based on $\Delta\theta$ for a deeper layer using the temperature profiles from the MWR measurements.

[Figure]

**Figure III.** Relationship between bulk Richardson number and potential temperature gradient.

*Changes to the manuscript*
The description of the coupling detection method (Sect 3.2) has been extended to better explain the method and the reasoning behind it.

- To better explain the use of the surface observations, the following text was added (new text is in bold):

  "**The premise of this criteria is that if the surface layer is stably stratified, the cloud must be decoupled from the surface as there exists a stable layer between the surface and cloud base. The θ-profile is used as a proxy for stability.** If the gradient of the 30 min mean θ between 2 and 10 m was positive **(e.g. an inversion was present between 2 and 10 m)**"

Figure 3: This figure is difficult to see. The authors may expand the figure to show the altitude range between 0 and 2 km and time period between 03:00-12:00.

We thank the referee for the feedback. We increased figure size and limited to the height and time period presented somewhat with the aim to increase the quality of the figure. However, we want to include also the cloud system present until 29 May 2018 01 UTC to make clear that there are clouds clearly identifiable as mixed-phase that were not included in our study.

*Changes in the manuscript:*
Modification of the Fig 2 (Fig 3 in the original manuscript) to increase figure clarity.

[Figure]

**Figure 2.** Example of the Cloudnet target classification product from 29 May, 18 UTC to 30 May 2018, 12 UTC. For the P-MPC, indicated by the gray dashed box, also the time series of coupling is shown. This case was classified as coupled.

L.315 (Fig.8a): The authors may compare the cloud base height with lifting condensation level (LCL) calculated from surface measurements.

Figure IV shows the lifting condensation level (LCL) calculated from the surface meteorological observations for all times when P-MPC were present and for each weather type (analogous to Fig. 8a in the original manuscript), as well as a comparison with the LCL and the cloud base height. The LCL was on average below 500 m, which is clearly lower the median cloud base height, which for P-MPC was 760 m (Sect. 4.2, L. 340). Similarly to cloud base height, the LCL was lower for westerly free tropospheric winds and higher for winds from N to SE. Comparing LCL to the cloud base (negative values indicate LCL below cloud base) underlines that cloud base height was usually above the LCL. There seems to be a dependency with wind direction, such that the P-MPC that were on average highest (weather types NE-SE, see Fig. 7a) are also the furthest away of the LCL. The P-MPC that were on average lowest (weather types SW-NW) were also closer to the LCL. This implies that the near surface air is further from condensation, either warmer or drier, for the easterly wind directions than westerly wind directions.

The lifting condensation level is defined as the height where condensation would occur would an air parcel be lifted from the surface adiabatically and without mixing with the surrounding air. However, when a surface based inversion is present such an air parcel would not be very likely to get lifted to such altitudes. Any inversion between the starting point and the LCL would indeed stop the journey of the air parcel. The discrepancy between the LCL and the cloud base height is another indication that the P-MPC were mostly decoupled from the surface. Although these results are somewhat interesting, we do not think the LCL is such a relevant parameter for the considered cloud regime and therefore chose not to include it in the manuscript.

[Figure]

**Figure IV.** Lifting condensation level (LCL) for each weather type (left) and the difference between the LCL and cloud base (right) for all P-MPC.

L.325: As far as I understood, the results described here are the most important one in this study. The authors may need to explain more on the physics behind. Higher LWP can be due to higher temperature (thermodynamic effect), moisture transport, lower stability (geometrically thicker clouds) and others. The authors may describe how these factors affect LWP under the westerly conditions, using IWV data and from viewpoint of seasonality.

The referee is commenting on the statement "the westerly weather types (SW, W and NW) were associated with lower P-MPCs and with more liquid and ice (mean LWP 42 gm$^{-2}$)". As stated above in the response for comment number 1, we have added the consideration of atmospheric humidity and temperature to Sect. 4.2. As a response for comment number 2, seasonality of LWP as well as atmospheric humidity and temperature were included in Sect. 4.3.

Section 4.3: In my opinion, this section can be moved to appendix or deleted. Figure 9 clearly shows that 850 hPa wind direction is more important than that at surface. The addition of surface wind analyses did not provide enough insights into the P-MPC.

We understand that the referee did not find the section in question as relevant as the other sections. While the local winds did not provide many additional insights for the P-MPC specifically, the wind climate of Kongsfjorden is one of the features that makes Ny-Ålesund different from many other Arctic sites with long term cloud observations. We therefore consider it worthwhile to address the local wind conditions. We would also like to point out that while these results might not be that interesting for the wider Arctic mixed-phase cloud community, they are of interest for those working with data obtained at Ny-Ålesund. However, we recognize the critique of the referee, and correspondingly have moved the parts detailing the variation in P-

MPC properties to an appendix and only shortly summarize the main findings. Additionally, the figure and related discussion about the relationship between coupling and surface wind direction was moved to this section to shift the focus on the result related to the surface winds that we found most interesting.

*Changes in the manuscript:*

- The section was moved to the end of Sect. 4, and is now Sect. 4.5 *Local wind patterns around Ny-Ålesund.*

- Figure 10, L. 354-367 and L. 471-488 of the original manuscript were moved to the appendix A1 *P-MPC properties*. These results are summarized in the main text (L. 544-550) as follows:

  "Regarding P-MPC properties, no strong relationships with surface wind direction were identified. Only the main findings are summarized here and further details are provided in Appendix A1. Considering weather types N, W and SW, which have the most cases across different surface wind directions, no statistically significant differences were found in the median liquid base height or cloud top temperature. The northwest surface wind was associated with the highest median LWP, possibly due to higher level of humidity available over the open sea. The southwest surface wind was associated with a significantly higher IWP for weather types W and SW (median IWP 16 gm$^{-2}$ and 18 gm$^{-2}$, respectively). However, these variations in LWP and IWP were not found for all three weather types analyzed." (L. 540-546)

- The related conclusion was re-formulated and now states: "Local winds were not found to impact the occurrence or the height of the P-MPCs, but for some free-tropospheric wind directions the surface wind direction was related to variations in LWP and IWP." (L. 582-584)

- Figure 11b, L. 385-386 and L. 503-507 of the original manuscript were moved to Sect. 4.5.

L.487: What is the "observed differences"?

We apologize the unclear writing. The "observed differences" simply referred to the differences presented in that section.

*Changes in the manuscript:*
"the observed differences in IWP and LWP" → "the differences found in IWP and LWP between different wind regimes" (L. 648-649)

L. 542: According to Nomokonova et al., (ACP2019), mean value of IWP of MPC was 164 gm$^{-2}$, while it was 12 gm$^{-2}$. Why the values are so different?

Nomokonova et al. (2019) included all profiles with a mixed-phase cloud layer, defined as both ice and liquid water identified in the same cloud layer. This includes amongst others deep and heavily precipitating cloud systems, clouds related with frontal passages and storms, and convective clouds associated with cold air outbreaks. Many of these clouds are mixed-phase and have a higher IWP than the persistent low-level MPC that are investigated in our study, which leads to a much higher mean IWP. For an example of a MPC not included in this work, see the cloud present 29 May 2018 18-01 UTC in Fig. 2.

L.544: Less and higher P-MPC-> Less frequent and higher cloud base height of P-MPC

Thank you for the better formulation.

*Changes in the manuscript:*

"Less and higher P-MPC" → "Less frequent P-MPC and with higher cloud base height" (L. 580)

L.554: The words "weather type" and "wind regime" are used in this study. Describe as "wind direction at 850 hPa" etc, such as in figure captions for Figure 4b.

*Changes in the manuscript:*
We changed the wording of the corresponding sentence:

"The variation of median LWP between different **weather types** (Fig. 8b) was larger than the variation found between different **wind regimes** (Fig. 10b) or coupling states (Fig. 12a)."

changed to

"The variation of median LWP between different **wind direction at 850 hPa** was larger than the variation found between different **surface wind regimes** or coupling states." (L. 587-589)

**Reply to Referee #2**

(1) Section 4 is mainly a description of statistics. No attempt to physically interpret these results is made before section 5. I would recommend to the authors to discuss the underlying physical processes and how these are supported by the statistics during section 4. It is hard to remember all the details of this section when reading the recap in section 5.

We thank the referee for this feedback and follow the recommendation.

*Changes in the manuscript:*
Sections 4 and 5 were merged and structured so that the relevant discussion is placed together with the presentation of the corresponding results. Sect. 4 is titled "*Results and discussion*" and now has thematic subsections.

(2) Seasonality is discussed in section 4.1 and regional wind patterns in section 4.2, but there is no attempt to investigate the links between these two factors. If a certain wind pattern dominates specific seasons, this should be considered when interpreting the results in section 4.2. The authors should present the frequency of the different wind patterns through the year.

The referee is correctly pointing out that the seasonality in persistent low-level mixed-phase cloud (P-MPC) occurrence together with a potential seasonal variation in regional wind direction should be considered when interpreting the results related to the connection between the P-MPC and wind direction. We have not found such seasonal variation in the weather type (used to describe the regional wind field at 850 hPa) that could explain the observed differences in cloud occurrence and properties between different weather types. To allow others to make the same conclusion, we have added a figure presenting the seasonal occurrence of all weather types (Fig. 10) in the manuscript.

*Changes in the manuscript:*
To clearly demonstrate that the dependency of P-MPC on weather types presented in Sect. 4.2 is not a result in seasonal variation of P-MPC occurrence and properties combined with a seasonality in 850 hPa wind direction, Fig. 10 is included in Sect. 4.3 with the following text:

"Since P-MPC properties (excluding LWP) as well as atmospheric temperature and humidity vary seasonally, a seasonal dependency in wind direction could explain the weather type dependent variations in P-MPC properties found in Sect. 4.2. To examine this possibility, Fig. 10 shows the proportion of P-MPC observations in each season for every weather type. The observation period of 2.5 years from June 2016 to October 2018 together with the seasonal variation in P-MPC occurrence (Fig. 5) lead to the uneven distribution of data between seasons. Overall, the summer months contribute most to the data set. However, there are no extensive differences found between the weather types. Most noteworthy is the high spring and low autumn occurrence of NW, which might contribute to the high IWP for this weather type (Fig. 7c). Furthermore, N and E were relatively more common in winter, N and SE more common in spring, and N less common in autumn. Given the lack of clear trends, we believe the seasonal variation in wind direction plays a minor role in the weather type dependent differences in P-MPC occurrence and properties described in the previous section." (L. 416-425)

[Figure]

**Figure 10.** Distribution of seasons in the studied data set for each weather type. Only time periods when a P-MPC was present are included to evaluate the possible impact of wind direction seasonality on cloud properties and occurrence.

(3) When using the MWR θ-profile to assess decoupling, Δθ between the surface and the height half way to the liquid layer is estimated. But if decoupling occurs between this level and liquid base height, then the algorithm would classify a decoupled cloud as coupled. Do results vary significantly when using the gradient between the surface and the level exactly below the liquid layer as criterion? Please check the uncertainty in the applied method.

We first elaborate on the reasoning behind the choice made, as this was not made clear enough in the manuscript, before replying to the explicit comments/questions made.

Figures 3a and b (see below) show two example cases, one for a coupled and one for a decoupled cloud. Comparing the θ-profiles from sounding and MWR highlights the challenge of using the MWR data: While the general shape of the profile can be retrieved, it is not possible to resolve sharp inversions or detailed structures of the profile. Temperature inversions are very common at cloud top, and impact the accuracy of the potential temperature retrieved from MWR measurements in the vicinity of the inversion. Examining the profiles at liquid base in Fig. 3 and b reveals that the MWR profile is deviating from the sounding profile already at and below liquid base. Figure 3c presents a comparison of MWR profiles with all available soundings when a P-MPC was present confirming this issue being present in most of the cases. At 0.5*liquid base height the impact of cloud top inversion is not present like it is at liquid base, which is why we chose this height to determine the stability of the subcloud layer.

The method does not explicitly identify cloud decoupling, instead the stability of the lower half of the subcloud layer is estimated and used as a proxy for decoupling. If decoupling takes place above this layer, it is common for the lower half of the subcloud layer to

[Figure]

**Figure V.** Distribution of the surface based mixing layer height in the sounding profiles classified as decoupled (165 cases)

be at least partially stable leading to a correct identification of decoupling. The reviewer states that if decoupling happens above the 0.5*liquid base height, the cloud would be erroneously considered coupled. This is only true if additionally the layer from surface up to this height is well-mixed. We analysed the soundings classified as decoupled to investigate the depth of a surface based mixing layer using the same method as for detecting the decoupling but from surface upwards. The distribution of the surface based " mixing layer height, normalized to cloud base height, is shown in Fig. V. The figure confirms the commonality of a stable lower half of the subcloud layer for decoupled clouds, which aids the MWR based algorithm in detecting decoupling.

The referee is further asking if results vary significantly if the potential temperature at liquid base height is used. Using potential temperature at 0.5*liquid base height yields 77 % of all MWR profiles classified as decoupled. If we use liquid base height instead, 95 % of all MWR profiles become decoupled. The difference is clear, and the agreement with soundings is worse when using θ at the liquid base height.

_Changes in the manuscript:_

- Fig 3a was replaced with two example profiles, one for a coupled and one for a decoupled case. The main outcome of the original Fig. 3a was already included in the text, but was slightly extended:

  "The resulting θ-profiles were compared with the profiles from radiosondes in the period June 2016– October 2018 (**Fig. 2a**). A slight cold bias is present, but in the lowest 2.5 km the RMSE is below 1.8 K"

  changed to

  "The resulting θ-profiles were compared with the profiles from radiosondes in the period June 2016– October 2018 (**not shown**). A slight cold bias is present **(< 0.4 K)**. **The RMSE increases with altitude**, but in the lowest 2.5 km the RMSE is **still** below 1.8 K." (L. 147-149)

[Figure]

**Figure 3.** Examples of θ-profiles from sounding and MWR, as well as surface observations: a coupled cloud on the 24 October 2017 (a) and a decoupled cloud on the 1 February 2018 (b). The blue shaded area indicates the cloud layer, where cloud base and top are determined as the median values of the Cloudnet based cloud base and top for the duration of the sounding. The gray dashed line indicates the decoupling height defined from the sounding θ-profile. Comparison of potential temperature profiles from sounding and retrieved from MWR measurements when P-MPC were present, with height normalized in respect to the liquid layer (c). Comparison of the diagnosed coupling with the new method based on MWR and surface observations and based on sounding profiles (d).

- To accompany the new figures (Fig 3. and b), the following text was added:
  "Fig. 3 a and b show two example cases, one for a coupled and one fora decoupled cloud, respectively. Both profiles demonstrate a structure typical for stratiform Arctic MPC a temperature inversion at cloud top, below which a well mixed layer is identifiable. In the case of the coupled P-MPC (Fig. 3a), the well-mixed layer extends to the surface. For the decoupled P-MPC (Fig. 3b) the well-mixed layer extends 200 m below the liquid layer base, below which several weaker temperature inversions and a generally stable stratification can be identified." (L. 190-194)

- The reasoning for the use of half of the liquid base height was extended, and is now:
  "The reason for using the height equalling half of the liquid layer base height can be understood by comparing the θ-profiles from sounding and MWR in Fig. 3 a and b. While the general shape of the profile can be retrieved from the MWR measurements, it is not possible to resolve sharp inversions or detailed structures of the profile. Yet temperature inversions are very common at the top of P-MPCs. The comparison of MWR profiles with all available soundings when a P-MPC was present (Fig. 3c) shows that the accuracy of the retrieved potential temperature is reduced in the vicinity of the liquid layer top and that the influence extends to below the liquid layer base. At 0.5*liquid base height the impact of cloud top inversion is smaller than at liquid base and the RMSE is below 1 K, which is why we chose this height to determine the stability of the subcloud layer. Note, that it should not be inferred that the method can only detect decoupling occurring in the lowest half of the subcloud layer. When decoupling occurs above the layer explicitly included, it is common that the lower half of the subcloud layer is at least partly stably stratified, as can also be seen in the example of Fig 3b, prompting a correct decoupling classification." (L. 206-216)

(4) A main conclusion is that cloud-surface coupling is more frequent when wind comes from the sea and that it enhances cloud liquid. However for these conditions coupling occurs for somewhat less than 50 % of the time. I suggest to the authors to investigate the meteorological conditions (e.g. large-scale moisture transport) between the decoupled and coupled cases when NW surface winds prevail. This might give indications of what drives coupling, which I don't think is the local wind pattern.

The referee is questioning two conclusions drawn from the analysis, firstly that the local wind pattern drives cloud-surface coupling, and secondly, that coupling with northwesterly surface wind enhances cloud liquid. We address both concerns individually.

Considering the relationship between surface coupling of P-MPC and the local wind patterns, we found that the frequency of coupling clearly varied between the different surface wind direction modes (Fig. 13b). This relationship was present in all seasons, although weaker in summer despite the seasonal variation in the surface wind (Fig. A2). However, we do not believe that the local wind patterns, approximated here by the surface wind direction, are the only factor determining the coupling of the cloud. The depth of the cloud driven mixing layer (which depends on both the generation of turbulence and the stratification of the adjacent layers) as well as the proximity of the cloud to the surface are both factors for the coupling of the cloud. However, based on our work and that of others, we believe there is evidence that the local winds associated with glacier outflows can act against the coupling of a cloud. In the literature the southeast and southwest surface winds have been associated with katabaticly driven flows from the nearby glaciers (Beine et al., 2001; Jocher et al., 2012; Argentini et al., 2003). Beine et al. argued that the northwest surface wind in summer is related to sea breeze. Argentini et al. evaluated the stability as a function of wind direction during the ARTIST campaign (15 March – 16 April 1998 at Ny-Ålesund). Fig. 5 in Argentini et al. 2003 (see below) shows that stable conditions were mainly observed with southeast surface wind and hardly ever with northwest wind. Unstable conditions occurred from 90º to 270º and under light wind conditions. Furthermore, they found large wind shear to generate turbulence and lead to neutral stratification. The

katabatic flows bring cold air down to the valley in a shallow layer close to the surface (Beine et al. 2001). In our opinion, such a cold surface layer would be efficient in decoupling the cloud above.

The hypothesis that glacier outflows act to decouple the P-MPC is based on the results in previous studies about modifications in the surface layer related to the local winds, and supported by the results we found in our study, namely Fig. 13b and A2. The manuscript in its original form required perhaps too much foreknowledge on the Kongsfjorden wind conditions for the argumentation to be clear. We have therefore extended the description of local wind conditions at Ny-Ålesund as well as the discussion related to these results to better explain our argumentation. Furthermore, we added the seasonality of the surface wind direction and associated coupling frequency in the appendix (Sect. A2) to provide a well rounded description of the surface wind conditions.

[Figure]

Argentini et al., 2003

**Fig. 5.** Histogram with the distribution of the observed cases for three different stability classes, unstable: $z/L < -0.05$; neutral: $-0.05 < z/L < 0.05$; stable: $z/L > 0.05$ during ARTIST.

For the second point related to the relationship between coupling and cloud liquid in P-MPC for northwesterly surface wind, we agree that the result was overemphasized. As pointed out by the referee, further analysis would be required to determine which processes are important for this subset of P-MPC cases. However, we find this additional work would be beyond the scope of this paper. While we still consider it worthwhile to point out the possible relationships between coupling, surface wind direction and amount of liquid in P-MPC, we have removed the corresponding statements from the conclusions and abstract of the paper.

*Changes in the manuscript*

- The description of the local wind climate in Sect. 3.4 was extended with a new paragraph summarizing previous studies:
  "The channeling of the free-tropospheric wind along the fjord axis is a typical feature of an Arctic fjord (Svendsen et al., 2002; Esau and Repina, 2012, and references therein). Previous work has found the feature prominent also at Kongsfjorden (Maturilli and Kayser, 2017). It is well documented that despite the dominating westerly free-tropospheric wind direction, in Kongsfjorden the near surface wind tends to blow southeasterly along the fjord axis (Maturilli and Kayser, 2017;

Beine et al., 2001; Jocher et al., 2012). This is usually attributed to katabatic forcing of the Kongsvegen glacier about 15 km east-southeast from Ny-Ålesund (Fig. 1), although Esau and Repina (2012) argued that for typical synoptic conditions the land-sea breeze circulation would be the dominant driver. The secondary mode in surface wind is from northwest, from the sea towards the island's interior. According to Jocher et al. (2012) the northwesterly surface winds are associated to cold air advection that relate to passing low-pressure systems. Beine et al. (2001) find this wind direction to be pronounced in June and July, which they associate with sea breeze and the melting of sea ice. In addition, at Ny-Ålesund weak southwesterly surface winds are observed, caused by katabatic flow from the Zeppelin mountain range and the Broggerbreen glacier south of Ny-Ålesund (Jocher et al., 2012; Beine et al., 2001) under specific synoptic conditions (Jocher et al., 2012; Argentini et al., 2003). The local wind conditions impact the stratification of the local boundary layer (Argentini et al., 2003; Svendsen et al., 2002). Argentini et al. show that during the ARTIST campaign (15 March – 16 April 1998 at Ny-Ålesund) stable conditions were mainly observed with southeast wind and hardly ever with northwest wind. Unstable conditions occurred from 90º to 270º and under light wind conditions. Furthermore, large wind shear was observed to generate turbulence and lead to neutral stratification. This brief summary of previous studies demonstrates the complexities of the local wind conditions present at the AWIPEV station." (L. 244-261)

- Because of the additional description of the local wind field based on literature, we removed some of the description of the wind conditions based on presented data. Specifically, the second example of the wind direction profiles, Fig. 5b of the original manuscript, was removed. The description of the wind profiles from the presented soundings was shortened. Figure 5a was combined with Fig. 4 and is Fig 5a has become Fig 4c in the revised manuscript.

- The discussion on the relationship between surface wind direction and coupling was extended to better explain the mechanisms in play:
  "Local winds in Kongsfjorden were quite apparently connected to the coupling of the P-MPC (Fig. 13b). Coupling was most common with northwest surface wind (from the sea) and least common with the southeast surface wind (towards the sea), and the same behaviour was found for every season despite the seasonality of both surface wind direction and cloud coupling (see Appendix A2). For the P-MPC to be thermodynamically decoupled from the surface, a stably stratified layer needs to exist between the surface and the cloud base. Argentini et al. report a dependence of surface layer stratification on wind direction (Argentini et al., 2003, Fig. 5). Stable conditions were most often found with southeast surface wind, for which only 8% of the P-MPC were considered coupled. On the other hand, stable conditions were rare with northwest surface wind, for which 37% of the P-MPC were coupled. The near surface wind from southwest and southeast is often related with flows from the glaciers (Jocher et al. 2012; Beine et al. 2001; Sect. 3.4) that bring cold air down to the valley in a shallow layer close to the surface. Such a cold surface layer is very efficient in decoupling the cloud and acts against the cloud driven turbulence that could otherwise couple the P-MPC to the surface. This effect might be stronger with southeast than southwest surface wind, since the katabatic winds from southwest are weaker (Fig. 4a). The differences in the coupling of the P-MPC with varying wind conditions can be explained by the differences in stratification of the lower boundary layer under different surface wind conditions. We conclude that the surface wind has the potential to modify the conditions in the boundary layer, which in turn can act to suppress coupling." (L. 547-561)

- Added Appendix A2 *Seasonality of surface wind direction and P-MPC coupling*, describing the seasonal variation in surface wind direction and the relationship with wind direction and P-MPC coupling. For complete text see the attached manuscript.

[Figure]

**Figure A2.** Wind rose for 30 min mean 10 m wind for each season (a-d) in the cloud observation period (June 2016--October 2018). The fraction of P-MPC cases classified as coupled, predominantly decoupled and fully decoupled for each surface wind direction mode in each season (e-f).

- Removed from conclusions: "Some of the observed differences between different wind regimes and coupling states might have been related (e.g. higher LWP were found for coupled P-MPC and for P-MPC associated with northwest surface wind, while coupling was most common for this surface wind direction)."

- Removed from abstract: "Furthermore, the near surface wind direction from the open sea was related to higher amounts of cloud liquid, and higher likelihood of coupling."

- Added at the end of Sect. 4.5 *Local wind patterns around Ny-Ålesund*: "There is a relationship between surface coupling and the local wind conditions at Ny-Ålesund, but to understand the impact of the combined effects on P-MPC properties would require further studies." (L. 569-570)

Abstract: it is stated that westerly clouds had a higher mean liquid (42 gm-2) and ice water path (16 gm$^{-2}$) compared to the overall mean of 35 and 12 gm-2, respectively. Is a 7 gm-2 difference in LWP important, given the large uncertainty in these retrievals? Moreover, I doubt that the impact on radiation is substantially different when changing cloud LWP by 7 gm-2. I don't think differences of this magnitude should be emphasized in the text.

The referee argues that the abstract mistakenly emphasizes the differences between P-MPC associated with westerly winds and all P-MPC. We agreed that this was not formulated correctly.

*Changes in the manuscript:*

"We found that persistent MPCs were most common with westerly winds, and the westerly clouds had a higher mean liquid (42 gm$^{-2}$) and ice water path (16 gm$^{-2}$) **compared to the overall mean of 35 and 12 gm$^{-2}$, respectively.**" (L. 8-9)

changed to

"We found that persistent low-level MPCs were most common with westerly winds, and the westerly clouds had a higher mean liquid (42 gm$^{-2}$) and ice water path (16 gm$^{-2}$) **compared to those with easterly winds.**" (L. 8-9)

Sections 2.3 and 3.3 offer a summary of the methods used to study the influence of the large-scale and the local wind patterns, respectively. However the first method is included in Section 2 (Observations) and the second in Section 3. It would make more sense if section 2.3 becomes a subsection of Section 3, too.

We agree, and thank for the suggestion.

*Changes in the manuscript:*
Section 2.3 *Circulation weather type* was moved to Sect. 3.

Section 4.1: The percentages given in this section would be more meaningful if the actual number of PMPCs and all-PMC cases included in the analysis is stated. Is it same as in Figure 2?

The criteria for identifying P-MPC is given in detail in Sect. 3.1. Because our interest was in persistent MPCs, one important case selection criteria was the condition for an liquid layer uninterrupted for at least 1h. The 'P-MPC case' is defined from the beginning to the end of the occurrence of the persistent liquid layer. The total number of these cases was 1412. The number given in Fig. 2 (Fig. 3 in the revised manuscript) refers to the number of sounding profiles that coincide with a P-MPC and when MWR data was available. The all-MPC occurrence refers to 30 second profiles that include co-existing liquid and ice. The total number of such profiles is 985901. Cases have not been identified for the all-MPC in the same sense as for the P-MPC. This is also mentioned in the text: "The 'all MPC' and the P-MPC occurrences in Fig. 5 are not directly comparable, since the first one refers to individual profiles and the latter is to a large extent defined by a temporally continuous liquid layer and also includes profiles without a mixed-phase layer detected" (L. 317-319).

*Changes in the manuscript:*
- Added in Sect. 4.1: "In total 1412 cases of P-MPC were identified." (L. 317)
- To clarify that the cloud occurrence statistics is based on the 30 s profiles, this is now mentioned when the results are introduced:
  "We first examine the frequency of occurrence of different types of clouds in the observation period of the cloud radar (10 June 2016–8 October 2018) **considering the 30 s averaged columns of the Cloudnet product.**" (L. 303-304)

L300-302: how different are the occurrence statistics in the sensitivity test?

Changing the longest allowed gap in the liquid layer from 5 to 2 min decreases the cases detected from 1412 to 1235. The fraction of P-MPC in the total data set decreases from 23% to 19%. Figure VI shows the monthly differences. Overall, using a stricter criteria shrinks the dataset.

[Figure]

**Figure VI.** Frequency of occurrence of P-MPC, allowing a maximum gap in the liquid layer to be 5 or 2 min in the P-MPC identification algorithm.

L481-484: here you combine Figure 5 and 10a to discuss how wind structure affects the PMPCs. However, Figure 5 corresponds to a much longer period than Figure 10a. For consistency, both wind and cloud measurements should correspond to the same time. Please check if utilizing fewer radiosondes results in very different wind structures. If this is the case, then you might consider removing/modifying the relevant discussion.

The referee expresses concerns that mistaken conclusions might be made because Fig. 5 includes the time period August 2011 – October 2018, when all other analysis were focused on the period June 2016 – October 2018. Here we show Fig. 4c (corresponding to Fig. 5a) that includes data only for the cloud observation period, June 2016 – October 2018. The original Fig. 5 is also included here (see below) for comparison. We appreciate the advice from the referee and for consistency changed the figure in the manuscript to only include the cloud observation period.

[Figure]

**Updated Fig. 5a** (Fig 4c in the revised manuscript) based on radiosoundings from June 2016 to October 2018.

*Changes in the manuscript:*
- The figure in question (Fig, 4c in the revised manuscript) was updated to only include soundings from the cloud observation period (June 2016 – October 2018).

[Figure]

Original Fig. 5 based on radiosoundings from August 2011 to October 2018.

Figure 6: it would be better if the actual number of profiles is also included in the figure.

The referee is suggesting to include the number of profiles in Fig. 5 of the revised manuscript. Unfortunately it is not possible to add the number of profiles in the figure, the only option would be to redo the figure using the number of profiles instead of relative frequencies. This is because the figure shows monthly values, and different months have different number of days, yielding to different number of profiles for the same frequency of occurrence in different months. Therefore we believe it would be misleading to use the number of profiles instead of the relative occurrence, as suggested by the referee.

Figure 9: the size **of** the dots

Done.

**References**

[revised manuscript text omitted]